# ROTATE: Regret-driven Open-ended Training for Ad Hoc Teamwork

## Abstract

Learning to collaborate with previously unseen partners is a fundamental generalization challenge in multi-agent learning, known as Ad Hoc Teamwork (AHT). Existing AHT approaches often adopt a two-stage pipeline, where first, a fixed population of teammates is generated with the idea that they should be representative of the teammates that will be seen at deployment time, and second, an AHT agent is trained to collaborate well with agents in the population. To date, the research community has focused on designing separate algorithms for each stage. This separation has led to algorithms that generate teammates with limited coverage of possible behaviors, and that ignore whether the generated teammates are easy to learn from for the AHT agent. Furthermore, algorithms for training AHT agents typically treat the set of training teammates as static, thus attempting to generalize to previously unseen partner agents without assuming any control over the set of training teammates. This paper presents a unified framework for AHT by reformulating the problem as an open-ended learning process between an AHT agent and an adversarial teammate generator. We introduce ROTATE, a regret-driven, open-ended training algorithm that alternates between improving the AHT agent and generating teammates that probe its deficiencies. Experiments across diverse two-player environments demonstrate that ROTATE significantly outperforms baselines at generalizing to an unseen set of evaluation teammates, thus establishing a new standard for robust and generalizable teamwork.

## 1 Introduction

As AI agents are deployed in diverse applications, it is increasingly crucial that they can collaborate effectively with previously unseen AI agents and humans. While methods for training teams of agents have been explored in cooperative multi-agent reinforcement learning (CMARL) (Foerster et al., 2018; Rashid et al., 2020), prior work highlighted that CMARL agents fail to perform optimally when collaborating with unfamiliar teammates (Vezhnevets et al., 2020; Rahman et al., 2021). Rather than learning strategies that are only effective against jointly trained teammates, dealing with previously unseen teammates requires adaptive AI agents that efficiently approximate the optimal strategy for collaborating with diverse teammates. The training of such adaptive agents has been explored within ad hoc teamwork (AHT) (Bowling & McCracken, 2005; Stone et al., 2010; Mirsky et al., 2022) and zero-shot coordination (ZSC) (Hu et al., 2020; Cui et al., 2021; Lupu et al., 2021).

Most work has decomposed AHT learning into two stages (Mirsky et al., 2022), consisting of first creating a fixed set of teammates, and then training an AHT agent using reinforcement learning (RL), based on interactions with teammates sampled from the set. Methods that focus on AHT agent learning typically rely on a human-designed heuristic-based or pretrained teammates (Papoudakis et al., 2021; Zintgraf et al., 2021; Rahman et al., 2021) and therefore struggle to handle novel behaviors outside the predefined set of teammates (Strouse et al., 2021; Carroll et al., 2019). Recent work enhances the generalization capabilities of AHT agent learning methods by substituting the predefined set of teammates with a generated collection of diverse teammates (Lupu et al., 2021; Rahman et al., 2024), which are trained to maximize different notions of diversity. One such diversity notion is *adversarial diversity* (Rahman et al., 2023; Charakorn et al., 2023), which seeks to generate a set of teams that cooperate well within teams, but not across teams. However, prior work (Cui et al., 2023; Sarkar et al., 2023; Charakorn et al., 2024) empirically demonstrates that adversarial diversity

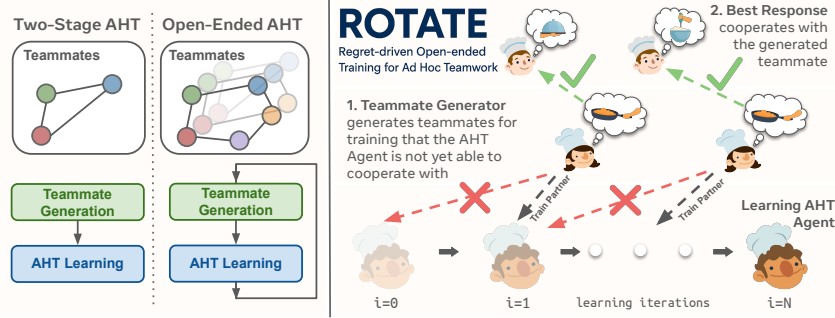

Figure 1: ROTATE *Overview.* ROTATE is an open-ended learning framework for AHT. The core idea of ROTATE is to improve the AHT agent by iteratively generating diverse teammates with whom the AHT agent struggles to collaborate, yet not so adversarial that effective teamwork becomes impossible.

often leads to teammate policies that actively diminish returns when interacting with agents other than their identified teammate, a phenomenon sometimes called *self-sabotage*.

This paper addresses two issues that cause current methods to fail to learn policies that effectively collaborate with some teammates. First, two-stage AHT methods (Papoudakis et al., 2021; Zintgraf et al., 2021; Rahman et al., 2021; 2023) learn from interacting with teammates from a small fixed training set. Even when the training set is diverse, the AHT agent remains incapable of collaborating effectively with some teammates sampled from the vast space of possible strategies, specifically those with significantly different behavior from the policies in the training set (Vezhnevets et al., 2020; Rahman et al., 2021). Second, other work designs a diverse training set of teammate policies by maximizing adversarial diversity (Charakorn et al., 2023; Rahman et al., 2023), which yields self-sabotaging teammates whose return-diminishing tendencies make it challenging for a randomly initialized RL-based AHT agent to learn to collaborate effectively (Cui et al., 2023; Sarkar et al., 2023). Despite addressing the first issue, some methods remain susceptible to the second issue by optimizing adversarial diversity (Yuan et al., 2023).

We address the shortcomings of using a small, fixed training set by proposing an open-ended learning framework that continually generates new teammates with whom the AHT agent interacts to enhance its collaborative capabilities. We formulate our learning objective by observing that maximizing the expected returns of an AHT agent on a known set of teammates is equivalent to minimizing its expected *cooperative regret*: the utility gap between the best response to a given teammate, and the AHT agent's performance with that teammate. While not knowing the teammates that will be encountered, we take inspiration from unsupervised environment design (UED) methods (Wang et al., 2020; Dennis et al., 2020; Jiang et al., 2021a; Rutherford et al., 2024a) and train an AHT agent to minimize its regret against generated teammates that maximize the AHT agent's cooperative regret. We propose a novel and practical objective that, unlike UED methods that optimize regret only at the initial state, also maximizes regret in states encountered later in an interaction. We build on these foundations to propose a practical algorithm, ROTATE (Fig. 1), which optimizes a cooperative regret-based minimax objective while maintaining a population of all teammates explored. We demonstrate that ROTATE significantly improves the robustness of AHT agents when faced with previously unseen teammates, compared to a range of baselines on two-player Level-Based Foraging and Overcooked tasks.

This paper makes three main contributions. First, it defines a novel problem formulation for AHT, enabling open-ended AHT training that continually generates new teammates. Second, it introduces a novel algorithm, ROTATE, that instantiates the proposed open-ended AHT framework. Third, it provides empirical evaluations demonstrating that ROTATE significantly improves return against unseen teammates compared to representative baselines from AHT and open-ended learning.

## 2 RELATED WORK

**Training AHT Agents.** The training of ego agent policies that near-optimally collaborate with diverse previously unseen teammates has been explored in AHT (Stone et al., 2010). Most AHT methods follow the two-stage design process, where the generation of a fixed training set of teammate policies is followed by AHT training. Given teammates from the training set, AHT methods (Mirsky et al., 2022) train an ego agent to model teammates (Albrecht & Stone, 2018) by first identifying their important characteristics (e.g., goals, beliefs, policies) based on their observed behavior, and

then estimating the best-response policy to these teammates based on the inferred characteristics. Recent AHT methods (Rahman et al., 2021; Papoudakis et al., 2021; Zintgraf et al., 2021; Wang et al., 2024a) use neural networks trained using reinforcement learning (Schulman et al., 2017; Mnih et al., 2016). To further improve AHT training, several approaches learn a distribution for sampling teammate policies during training based on maximizing the worst-case returns (Villin et al., 2025) or regret (Erlebach & Cook, 2024; Chaudhary et al., 2025) of trained agents. While few, exceptions to the two-stage process include methods designed for continual AHT (Nekoei et al., 2021; 2023; Yuan et al., 2023), methods that co-evolve populations of ego agents and teammates (Xue et al., 2025; Yuan et al., 2023), self-play based methods, which do not explicitly optimize for diversity (Yan et al., 2023; Cornelisse & Vinitsky, 2024), and empirical game theoretic methods that optimize for cooperative diversity as a heuristic to induce generalization to unseen teammates (Li et al., 2023).

**Teammate Generation for AHT & ZSC.** Recent work removes the need to predefine teammate policy sets by generating diverse teammates during or before agent training. Other-Play (Hu et al., 2020) creates symmetry-equivalent teammates while training the agent policy, while E3T (Yan et al., 2023) mixes the agent's current policy with a random policy to encourage diversity. FCP (Strouse et al., 2021) trains teammates via repeated CMARL runs with different seeds, later improved by methods maximizing information-theoretic diversity objectives such as Jensen-Shannon divergence (Lupu et al., 2021), mutual information (Lucas & Allen, 2022), and entropy (Xing et al., 2021; Zhao et al., 2023). More recent approaches (Charakorn et al., 2023; Rahman et al., 2024; Yuan et al., 2023) generate teammates that require distinct best-response strategies by maximizing adversarial diversity metrics, similar to ROTATE's cooperative regret. Unlike ROTATE, these methods (i) maximize regret between generated teammates rather than with the trained agent, (ii) fix the teammate set prior to training, and (iii) evaluate regret only at the initial state. This last property leads to sabotaging teammates that harm cooperation in states unseen in self-play, motivating heuristic solutions in prior work (Cui et al., 2023; Sarkar et al., 2023), and a systematic objective in ROTATE.

**Open-Ended Learning (OEL).** OEL (Langdon, 2005; Taylor, 2019) studies algorithms that continually generate novel tasks to train generally capable agents (Hughes et al., 2024; Baker et al., 2019). Many OEL approaches in RL take the form of unsupervised environment design (UED) (Dennis et al., 2020), which improves generalization by designing or sampling new environments with varied initial states. Some methods directly train neural networks to propose environments that induce high regret in the agent (Dennis et al., 2020), while others selectively sample curated tasks generated by procedural generators based on criteria such as expected return (Wang et al., 2020), TD-error (Jiang et al., 2021b), regret (Jiang et al., 2021a), or learnability (Rutherford et al., 2024a). In competitive MARL, OEL often produces new opponents through self-play (Silver et al., 2016; Lin et al., 2023). For AHT, MACOP (Yuan et al., 2023) generates novel teammates via an adversarial diversity objective optimized with evolutionary methods and similar to the objectives studied by by Charakorn et al. (2023) and Rahman et al. (2023). Thus, the objective can yield sabotaging teammates when applied only to the initial state. In contrast, ROTATE adopts a more systematic training objective that we demonstrate leads to performance gains.

## 3 BACKGROUND

The interaction between agents in an AHT setting may be modeled as a decentralized Markov decision process (Dec-MDP) (Bernstein et al., 2002). A Dec-MDP is characterized by a 7-tuple, $\langle N, S, \{\mathcal{A}^i\}_{i=1}^{|N|}, P, p_0, R, \gamma \rangle$, where $N$, $S$, and $\gamma$ respectively denote the index set of agents within an interaction, the state space, and a discount rate, $0 \leq \gamma \leq 1$. Every interaction between agents begins from a state sampled from the initial state distribution, $s_0 \sim p_0(s)$. At timestep $t$, each agent, $i \in N$, jointly executes an action selected from its action space, $a_t^i \in \mathcal{A}^i$, based on the observed state, $s_t$, and its policy, $\pi^i(s_t^i)$. We assume that teammates choose their actions only based on the current state. Meanwhile, the AHT agent, also referred to as the *ego agent*, selects actions based on its state-action history, which is necessary to distinguish between different types of teammates effectively. Denoting the set of all probability distributions over a set $X$ as $\Delta(X)$, the execution of the joint action, $a_t = (a_t^1, \ldots, a_t^{|N|})$, results in agents observing a new state, $s_{t+1}$, sampled according to the environment transition function, $P : S \times \mathcal{A}^1 \times \cdots \times \mathcal{A}^{|N|} \mapsto \Delta(S)$, and a common scalar reward, $r_t$, based on the reward function, $R : S \times \mathcal{A}^1 \times \cdots \times \mathcal{A}^{|N|} \mapsto \mathbb{R}$.

## 4 AD HOC TEAMWORK PROBLEM FORMULATION

AHT methods aim to train an adaptive policy that an ego agent can follow to achieve maximal return when collaborating with an unknown set of evaluation teammates. Formalizing the interaction between agents as a Dec-MDP, this section outlines the objective of AHT. While the most general AHT setting considers a possibly varying number of ego agents and teammates within an interaction (Wang et al., 2024a; Rahman et al., 2021), this formalization addresses the more straightforward case where there is only a single ego agent within a team.

Let $\pi^{\text{ego}}$ refer to the ego agent's policy, and $\pi^{-i}$ denote the $|N| - 1$ policies of the AHT agent's teammates. We denote the returns of an ego agent that follows $\pi^{\text{ego}}$ to collaborate with teammates controlled by $\pi^{-i}$, starting from state $s$, as:

$$V(s|\pi^{-i}, \pi^{\text{ego}}) = \mathbb{E}_{\substack{a_t^{\text{ego}} \sim \pi^{\text{ego}}, \\ a_t^{-i} \sim \pi^{-i}, P}} \left[ \sum_{t=0}^{\infty} \gamma^t R(s_t, a_t) \Big| s_0 = s \right]. \tag{1}$$

Let $\Pi^{\text{eval}}$ denote the unknown set of joint teammate policies encountered during evaluation, which is assumed to only contain competent and non-adversarial policies, as defined in the seminal work of Stone et al. (2010). Let $\psi^{\text{eval}}(\Pi^{\text{eval}})$ denote the probability distribution over $\Pi^{\text{eval}}$ defining how teammates are sampled during evaluation. An ego agent policy, $\pi^{\text{ego}}$, is evaluated by its ability to maximize the expected returns when collaborating with joint teammate policies sampled from $\psi^{\text{eval}}(\Pi^{\text{eval}})$, which is formalized as:

$$\max_{\pi^{\text{ego}}} V(\psi^{\text{eval}}, \Pi^{\text{eval}}, \pi^{\text{ego}}) = \max_{\pi^{\text{ego}}} \mathbb{E}_{\pi^{-i} \sim \psi^{\text{eval}}(\Pi^{\text{eval}}), s_0 \sim p_0} \left[ V(s_0|\pi^{-i}, \pi^{\text{ego}}) \right]. \tag{2}$$

An optimal $\pi^{\text{ego}}$ that maximizes Eq. 2 closely approximates the *best response policy* performance when collaborating with $\pi^{-i} \in \Pi^{\text{eval}}$. Given a teammate policy $\pi^{-i}$, $\text{BR}(\pi^{-i})$ is a best response policy to $\pi^{-i}$ if and only the team policy formed by $\pi^{-i}$ and $\text{BR}(\pi^{-i})$ achieves maximal return:

$$\text{BR}(\pi^{-i}) \in \arg\max_{\pi} \mathbb{E}_{s \sim p_0} \left[ V(s|\pi, \pi^{-i}) \right]. \tag{3}$$

In some cases, AHT algorithms can estimate this optimal policy by using $\Pi^{\text{eval}}$ to train an ego agent policy that maximizes $V(\psi^{\text{eval}}, \Pi^{\text{eval}}, \pi^{\text{ego}})$ when $\Pi^{\text{eval}}$ is known.[1] However, most AHT methods address the more challenging case where $\Pi^{\text{eval}}$ is unknown, which is the setting that this paper adopts as well. While our methods assume no knowledge of $\Pi^{\text{eval}}$ during training, we follow standard practice (Papoudakis et al., 2021; Rahman et al., 2021; Zintgraf et al., 2021; Wang et al., 2024a) by manually designing a diverse $\Pi^{\text{eval}}$ for evaluation purposes, as we later describe in Section 7.

When $\Pi^{\text{eval}}$ is unknown, AHT algorithms (Mirsky et al., 2022) learn by interacting with policies from the training set, $\Pi^{\text{train}}$, which are learned or manually designed by leveraging an expert's domain knowledge about the characteristics of $\Pi^{\text{eval}}$. After forming the set of training teammates, current AHT algorithms use RL to discover an ego agent policy based on interactions with joint policies sampled from $\Pi^{\text{train}}$. While the precise training objective varies with the AHT algorithm, methods commonly estimate the ego agent policy maximizing the expected return during interactions with joint policies sampled uniformly from $\Pi^{\text{train}}$, which we describe below:

$$\pi^{*,\text{ego}}(\Pi^{\text{train}}) = \arg\max_{\pi^{\text{ego}}} \mathbb{E}_{\pi^{-i} \sim \mathcal{U}(\Pi^{\text{train}}), s_0 \sim p_0} \left[ V(s_0|\pi^{-i}, \pi^{\text{ego}}) \right]. \tag{4}$$

Naturally, even $\pi^{*,\text{ego}}(\Pi^{\text{train}})$ may be suboptimal with respect to $\Pi^{\text{eval}}$ and $\psi^{\text{eval}}$, due to the potential distribution shift caused by differences between the training and evaluation objectives.

## 5 REFORMULATING AD HOC TEAMWORK AS AN OPEN-ENDED LEARNING PROBLEM

In this section, we show how the idealized ad hoc teamwork objective—training ego agents to collaborate well with unknown teammates (Eq. 2, Section 4)—can be operationalized as a cooperative

---

[1]In the context of reinforcement-learning-based AHT algorithms, "known" means that an AHT algorithm has unlimited sampling access to the teammate policies.

regret-driven, open-ended learning procedure. In particular, we show that for a fixed set of teammates $\Pi^{\text{eval}}$ and sampling distribution $\psi^{\text{eval}}$ over $\Pi^{\text{eval}}$, maximizing the return of the ego agent is equivalent to minimizing its cooperative regret. In absence of knowledge about $\Pi^{\text{eval}}$ and $\psi^{\text{eval}}$, we argue that minimizing the *worst-case* cooperative regret of the ego agent with respect to regret maximizing teammates leads to ego agents that cooperate well with any unknown teammate. Based on this, we propose a novel *minimax regret* objective (Eq. 7). Finally, we present an algorithmic framework for optimizing the minimax regret objective in an iterative fashion (Alg. 1).

We define the *cooperative regret* of an ego agent policy $\pi^{\text{ego}}$ when interacting with some joint teammate policy $\pi^{-i}$ from a starting state $s$ as:

$$\text{CR}(\pi^{\text{ego}}, \pi^{-i}, s) = V\left(s|\pi^{-i},\ \text{BR}(\pi^{-i})\right) - V\left(s|\pi^{-i}, \pi^{\text{ego}}\right). \tag{5}$$

Any optimal AHT policy that maximizes Eq. 2 must also minimize the expected regret over joint teammate policies sampled based on $\psi^{\text{eval}}(\Pi^{\text{eval}})$, which we formally express as:

$$\text{CR}(\psi^{\text{eval}}, \Pi^{\text{eval}}, \pi^{\text{ego}}) = \mathbb{E}_{\pi^{-i} \sim \psi^{\text{eval}}(\Pi^{\text{eval}}), s_0 \sim p_0}\left[\text{CR}(\pi^{\text{ego}}, \pi^{-i}, s_0)\right]. \tag{6}$$

This property is a consequence of $V\left(s|\pi^{-i}, \text{BR}(\pi^{-i})\right)$ being independent of $\pi^{\text{ego}}$ for any $\pi^{-i}$ and $s$, leaving maximizing expected regret equivalent to minimizing the negative expected returns when collaborating with joint teammate policies sampled from $\psi^{\text{eval}}(\Pi^{\text{eval}})$.

Without knowing $\Pi^{\text{eval}}$ to optimize $\text{CR}(\psi^{\text{eval}}, \Pi^{\text{eval}}, \pi^{\text{ego}})$, we instead take inspiration from approaches in UED (Wang et al., 2020; Dennis et al., 2020), and propose optimizing $\pi^{\text{ego}}$ to minimize the *worst-case regret* that could be induced by any teammate policy $\pi^{-i}$:

$$\min_{\pi^{\text{ego}}} \max_{\pi^{-i} \in \Pi^{-i}} \mathbb{E}_{s_0 \sim p_0}\left[\text{CR}(\pi^{\text{ego}}, \pi^{-i}, s_0)\right], \tag{7}$$

where $\Pi^{-i}$ denotes the set of all competent and non-adversarial (Stone et al., 2010) joint teammate policies. Limiting the considered joint policies is important, as teams that consistently perform poorly against any $\pi^{\text{ego}}$ are unlikely to be encountered in coordination scenarios and may introduce unnecessary learning challenges for RL-based AHT learning algorithms.

Finding $\pi^{\text{ego}}$ that achieves zero worst-case regret is equivalent to finding an ego agent that achieves the best-response return with any joint teammate policy $\pi^{-i}$. If such a $\pi^{\text{ego}}$ exists, then this AHT agent would maximize Eq. 2 for any $\psi^{\text{eval}}$ and $\Pi^{\text{eval}}$. However, its existence is not guaranteed (Loftin & Oliehoek, 2022). In practice, we are content with *minimizing* the worst-case regret. While minimizing worst-case regret still applies to AHT problems with more than one teammate, we limit our method for optimizing Eq. 7 and our experiments to two-player, fully observable AHT games.

---

**Algorithm 1** Open-Ended Ad Hoc Teamwork Framework

---

**Require:**
    Environment, Env.
    Total of training iterations, $T^{\text{iter}}$.
    Initial ego agent policy parameters, $\theta^{\text{ego}}$.
1:  $\text{B}_\pi \leftarrow \langle\rangle$                                                  ▷ Init teammate policy parameter buffer.
2:  **for** $j = 1, 2, \dots, T^{\text{iter}}$ **do**
3:      $\text{B}_\pi^{\text{new}} \leftarrow$ **TeammateGenerator**(Env, $\theta^{\text{ego}}$, $\text{B}_\pi$)    ▷ Train teammates to maximize regret.
4:      $\theta^{\text{ego}} \leftarrow$ **EgoUpdate**(Env, $\theta^{\text{ego}}$, $\text{B}_\pi^{\text{new}}$)         ▷ Train ego agent to minimize regret.
5:      $\text{B}_\pi \leftarrow \text{B}_\pi^{\text{new}}$
6:  **end for**
7:  **Return** $\theta^{\text{ego}}$

---

Algorithm 1 outlines our general framework for training an ego agent to minimize the worst-case cooperative regret induced by any teammate $\pi^{-i} \in \Pi^{-i}$. Algorithm 1 resembles coordinate ascent algorithms (d'Esopo, 1959), which alternate between optimizing for $\pi^{-i}$ and $\pi^{\text{ego}}$ for $T^{\text{iter}}$ iterations, while assuming the other is fixed. We call a phase where we fix $\pi^{\text{ego}}$ and update $\pi^{-i}$ to maximize the ego agent's regret, the *teammate generation phase*. Meanwhile, assuming that $\pi^{-i}$ is fixed, the *ego agent update phase* updates $\pi^{\text{ego}}$ to minimize regret.

Our practical algorithm, ROTATE, instantiates Algorithm 1 by specifying the **TeammateGenerator** and **EgoUpdate** procedures, and is described in Section 6. It is an open-ended learning procedure

according to the definition proposed by Hughes et al. (2024), because it continually generates *novel* yet *learnable* artifacts (i.e., teammates) for an observer (i.e., ego agent). A discussion of how ROTATE satisfies the definition of Hughes et al. (2024) is provided in App. C.1.

# 6 PRACTICAL ALGORITHM: ROTATE

This section presents our practical algorithm for optimizing the minimax regret objective proposed in Section 5, ROTATE. We first describe the teammate generation procedure in Section 6.1, focusing on motivating the objective used to generate teammate policies. Next, we describe the ego agent update method in Section 6.2. App. A provides the ROTATE pseudocode and a more detailed discussion of the losses and exact update procedure.

## 6.1 ROTATE TEAMMATE GENERATOR

Given a fixed $\pi^{\text{ego}}$, ROTATE's teammate generator seeks to discover a teammate policy that maximizes the cooperative regret of $\pi^{\text{ego}}$. Maximizing cooperative regret requires estimating the teammate policy, $\pi^{-i}$, and its best response policy, $\text{BR}(\pi^{-i})$. In the following, we abbreviate $\text{BR}(\pi^{-i})$ to BR for brevity. ROTATE's teammate generator estimates both policies using the Proximal Policy Optimization (PPO) algorithm (Schulman et al., 2017).

The *per-trajectory regret* of $\pi^{\text{ego}}$ (i.e., the inner objective of Eq. 7) is the regret from trajectories starting from the initial state distribution:

$$\max_{\pi^{-i}} \mathbb{E}_{s_0 \sim p_0} \left[ \text{CR}(\pi^{\text{ego}}, \pi^{-i}, s_0) \right]. \tag{8}$$

Eq. 8 resembles the objectives used in past UED (Wang et al., 2020; Dennis et al., 2020) and the teammate generation literature (Rahman et al., 2024; Charakorn et al., 2023) to generate tasks or teammate policies. Recent work demonstrates that maximizing per-trajectory regret is prone to yielding self-sabotaging teammates (Cui et al., 2023; Sarkar et al., 2023). Maximizing the cooperative regret only from $s_0 \sim p_0$ implicitly encourages $\text{BR}(\pi^{-i})$ to select actions leading to future states that are distinguishable from those encountered during the interaction between $\pi^{-i}$ and $\pi^{\text{ego}}$. When encountering future states from interactions with $\pi^{\text{ego}}$, $\pi^{-i}$ ends up choosing actions that sabotage cooperation by minimizing the teammate's returns against $\pi^{-i}$. Thus, training $\pi^{\text{ego}}$ to minimize regret (i.e., by maximizing the expected returns) when collaborating with $\pi^{-i}$ using RL becomes challenging because $\pi^{-i}$ actively chooses actions that undermine collaboration.

We mitigate the emergence of self-sabotage by training $\pi^{-i}$ to maximize two objectives across states sampled from different state visitation distributions. These state visitation distributions result from: (i) teammate and BR interactions (self-play, SP), (ii) teammate and ego agent interactions (cross-play, XP), and (iii) cross-play continued by self-play interactions (SXP), where the teammate is first interacting with the ego agent, but switches at a random timestep $t$ to interacting with its BR. Let $d(\pi^1, \pi^2; p)$ denote the state visitation distribution when $\pi^1$ and $\pi^2$ interact based on a starting state distribution $p$. We use the following shorthand to denote the SP, XP, and SXP state visitation distributions:

$$p_{\text{SP}} := d\left(\pi^{-i}, \text{BR}(\pi^{-i}); p_0\right), \;\; p_{\text{XP}} := d\left(\pi^{-i}, \pi^{\text{ego}}; p_0\right), \;\; p_{\text{SXP}} := d\left(\pi^{-i}, \text{BR}(\pi^{-i}); p_{\text{XP}}\right). \tag{9}$$

Based on these distributions, we define the following *per-state regret* objective for training $\pi^{-i}$:

$$\max_{\pi^{-i}} \left( \mathbb{E}_{s \sim 0.5\, p_{\text{XP}} + 0.5\, p_{\text{SP}}} \left[ \text{CR}(\pi^{\text{ego}}, \pi^{-i}, s) \right] + \mathbb{E}_{s \sim p_{\text{SXP}}} \left[ V(s | \pi^{-i}, \text{BR}(\pi^{-i})) \right] \right). \tag{10}$$

The difference between the per-trajectory and per-step regret objectives is visualized in Figure 2. Both terms in the per-state regret objective discourage adversarial behavior from $\pi^{-i}$. The first term in Expr. 10 corresponds to the ego agent's regret starting from both SP and XP states. Estimating regret from XP and SP requires collecting SXP data as well as an analogous type of data called XSP (SP continued by XP interactions), as detailed in App. A.2. In general, optimizing the ego agent's regret encourages discovering $\pi^{-i}$ for which the ego agent policy has a high room for improvement. Optimizing regret starting from XP states requires $\pi^{-i}$ to be able to coordinate with its BR starting from any state encountered during interactions with the ego agent, thus preventing $\pi^{-i}$ from irrecoverably sabotaging an interaction. On the other hand, optimizing regret from SP states requires $\pi^{-i}$ to be able to decrease the return of the ego agent starting from any state encountered

during interactions between the teammate and the BR, thus disincentivizing the emergence of unconditional cooperation signals. Finally, we find that training $\pi^{-i}$ to collaborate well with its BR even during SXP interactions helps ensure that $\pi^{-i}$ is a good-faith collaborator with at least one partner.

While obtaining states from $p_{\text{SP}}$ and $p_{\text{XP}}$ is straightforward, states from $p_{\text{SXP}}$ and $p_{\text{XSP}}$ are collected using either environment resetting or policy switching. Using SXP as an example, if an environment supports resetting to any arbitrary state, then states encountered during XP interaction can be stored and used as the initial state for SP interactions. Otherwise, we may sample a random timestep $t$, run XP interaction until timestep $t$, and then switch to SP interaction (Sarkar et al., 2023). Only data gathered after timestep $t$ should be used to compute objectives based on $p_{\text{SXP}}$.

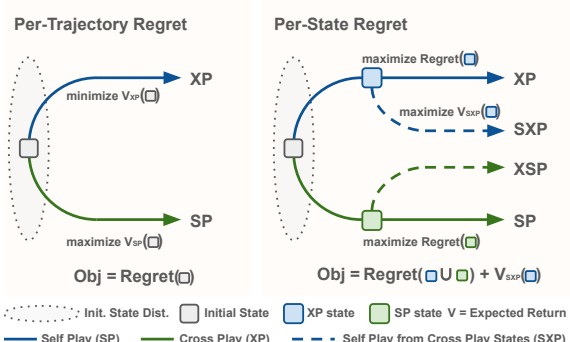

Figure 2: Teammate policy optimization objectives: per-trajectory regret vs per-state regret.

### 6.2 ROTATE Ego Agent Update

At each iteration, ROTATE creates a teammate that attempts to discover cooperative weaknesses of the previous iteration's ego agent, by maximizing its per-state regret. To allow the ROTATE ego agent to improve its robustness over time and reduce the possibility that it forgets how to cope with previously generated teammates, the ROTATE ego agent maintains a *population buffer* of generated teammates. During the ego agent update phase of each iteration, the ROTATE ego agent is trained using PPO (Schulman et al., 2017) against teammates sampled uniformly from the population buffer. We find experimentally that for the ego agent to learn effectively against the nonstationary population buffer, it is important to define a lower entropy coefficient and learning rate than when training the teammate and BR agents (typically in the range of $1 \times 10^{-4}$ for the entropy coefficient and $1 \times 10^{-5}$ for the learning rate).

## 7 Experimental Results

This section presents the empirical evaluation of ROTATE compared to baseline methods, as well as several ablations. The experiements consider one illustrative matrix game and six benchmark tasks. Supplemental results, implementation details, and code link are provided in the Appendix. The main research questions are:

- **RQ1**: Does ROTATE better generalize to unseen teammates, compared to baseline methods from the AHT and UED literature? (Yes)
- **RQ2**: Does per-state regret mitigate sabotage and improve generalization to unseen teammates compared to per-trajectory regret? (Yes)
- **RQ3**: Does the SXP return term of the ROTATE teammate generation objective improve learning and generalization? (Yes)
- **RQ4**: Is the population buffer necessary for ROTATE to learn well? (Yes)

### 7.1 Experimental Setup

This section describes the experimental setting, including tasks, baselines, construction of the evaluation set, and the evaluation metric.

**Tasks** ROTATE is evaluated on a didactic matrix game and six benchmark tasks. For clarity, the matrix game is described with the corresponding results. The benchmark tasks are Level-Based Foraging (LBF) (Albrecht & Ramamoorthy, 2013), and the five classic layouts from the Overcooked suite (Carroll et al., 2019): Cramped Room (CR), Asymmetric Advantages (AA), Counter Circuit (CC), Coordination Ring (CoR), and Forced Coordination (FC). All six tasks are cooperative, permit a variety of possible conventions, and are commonly used within the AHT literature (Albrecht & Ramamoorthy, 2013; Christianos et al., 2020; Papoudakis et al., 2021). In LBF, two agents must navigate to apples that are randomly placed within a gridworld, and cooperate to pick up the apples.

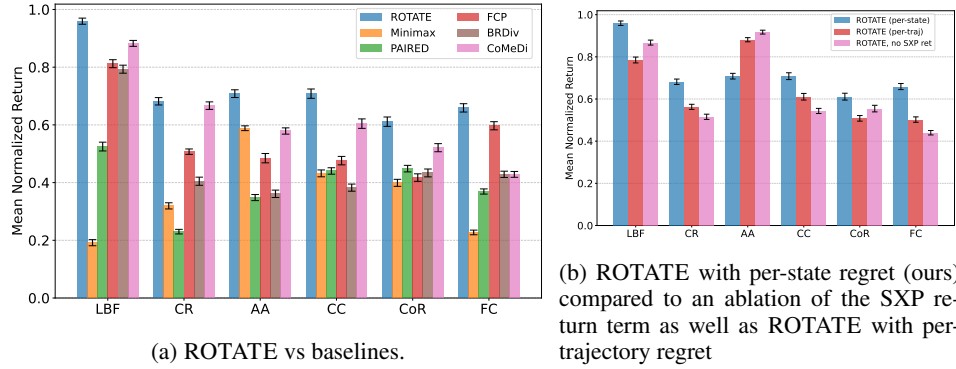

(a) ROTATE vs baselines.

(b) ROTATE with per-state regret (ours) compared to an ablation of the SXP return term as well as ROTATE with per-trajectory regret

Figure 3: (Left) ROTATE outperforms all baseline methods across all tasks in evaluation return. (Right) ROTATE with per-state regret (ours) outperforms ROTATE with per-trajectory regret in 5/6 tasks. 95% bootstrapped CI's are shown, computed across all evaluation teammates and trials.

In all Overcooked tasks, two agents collaborate in varying gridworld kitchen layouts to prepare dishes. All experiments were implemented with JAX (Bradbury et al., 2018).

**Baselines** As our method is most closely related to methods from UED and teammate generation, we compare against two UED methods adapted for AHT (PAIRED (Dennis et al., 2020), Minimax Return (Morimoto & Doya, 2005; Villin et al., 2025)) and three teammate generation methods (Fictitious Co-Play (Strouse et al., 2021), BRDiv (Rahman et al., 2023), CoMeDi (Sarkar et al., 2023)). While curator-based methods such as PLR (Jiang et al., 2021a;b) are prevalent in UED, we do not compare against them as they are orthogonal to ROTATE (Erlebach & Cook, 2024; Villin et al., 2025; Chaudhary et al., 2025). Similarly, we do not compare against AHT algorithms for ego learning (Albrecht & Stone, 2018). Each baseline is described in detail in App. B. For fair comparison, all open-ended and UED methods were trained for a similar number of environment interactions, or until best performance on the evaluation set. All teammate generation approaches were ran using a similar number of environment interactions as their original implementations, as scaling them up to use a similar number of steps as the open-ended approaches proved challenging (see discussion in App. B). All results are reported with three trials.

**Construction of $\Pi^{\text{eval}}$** We wish to evaluate all methods on as diverse a set of evaluation teammates as practically feasible, while ensuring that each teammate acts in "good faith". To achieve this goal, for each task, we construct 9 to 13 evaluation teammates using three methods: IPPO with varied seeds and reward shaping, BRDiv, and manually programmed heuristic agents. Full descriptions of the teammate generation procedure and all teammates in $\Pi^{\text{eval}}$ are provided in App. G.

**Evaluation Metric** Ego agent policies are evaluated with each teammate in $\Pi^{\text{eval}}$ for 64 evaluation episodes, where the return is computed for each episode, and normalized using a lower return bound of zero and an estimated best response return as the upper bound for each teammate. Performance of a method on $\Pi^{\text{eval}}$ is reported as the normalized mean return with bootstrapped 95% confidence intervals, computed via the `rliable` library (Agarwal et al., 2021). Our normalized return metric is similar to the BRProx metric recommended by Wang et al. (2024b). Details about the normalization procedure and specific bounds for each teammate are reported in the App. G.

## 7.2 RESULTS

This section addresses the research questions introduced at the beginning of Section 7. Supplemental analysis considering alternative regret-based objectives, independent utility of the ROTATE population, performance breakdowns by evaluation teammate, learning curves for all variants of ROTATE, and a human proxy evaluation on Overcooked, are provided in App. D.

**RQ1: Does ROTATE better generalize to unseen teammates compared to baselines? (Yes)**
We evaluate ROTATE's generalization capabilities by comparing its performance against baselines on $\Pi^{\text{eval}}$. Fig. 3a compares the normalized mean returns for ROTATE and baseline methods across the six tasks. The results show that ROTATE significantly outperforms all baselines on 5/6 tasks.

Among the baseline methods, the next best performing baselines are CoMeDi and FCP. We attribute CoMeDi's strong performance to the resemblance of its mixed-play objective to our per-state regret objective, which we discuss in App. C.3. FCP's performance may be attributed to the large number of partners that FCP was trained with (approximately 100 teammates per task). We found that FCP tends to perform especially well with the IPPO policies in $\Pi^{\text{eval}}$, likely because the IPPO evaluation teammates are in-distribution for the distribution of teammates constructed by FCP.

Minimax Return performs surprisingly well in AA, which may be attributed to AA's particular characteristics. In AA, agents operate in separated kitchen halves, possessing all necessary resources for individual task completion, with pots on the dividing counter being the only shared resource. A team where both agents act fully independently may achieve high returns–albeit coordination leads to still higher returns.[2] Visualizing the trained policies reveals that the adversarial teammate trained by Minimax Return cannot drive the ego agent's return to zero, and does not prevent the ego agent from learning how to perform the task independently. However, on LBF and FC, where coordination is crucial to obtain positive returns, Minimax Return is the worst-performing baseline.

BRDiv and PAIRED exhibit comparatively poor performance, which may be partially attributed to their teammate generation objectives that resemble per-trajectory regret. As we find for **RQ2**, per-state regret outperforms per-trajectory regret within the ROTATE framework. Furthermore, PAIRED's update structure involves a lockstep training process for the teammate generator, best response, and ego agent. This synchronized training may hinder the natural emergence of robust conventions that are crucial for effective AHT.

|   | $H$ | $T$ | $S$ |
|---|---|---|---|
| $H$ | 1 | 0 | -1 |
| $T$ | 0 | 1 | -1 |
| $S$ | -1 | -1 | -1 |

Table 1: Payoff matrix for the sabotage game.

**RQ2a: Does per-state regret mitigate sabotage compared to per-trajectory regret? (Yes)** We design a simple *sabotage game* to investigate the whether the per-state regret objective leads to teammate policies that sabotage less often compared to the per-trajectory regret objective. The sabotage game is a fully cooperative, iterated matrix game with payoff matrix shown in Table 1. Each agent observes a game state that consists of the complete history of joint actions. Both agents have three actions: H, T, and S(abotage). The first two actions lead to two possible cooperative outcomes, while the last action leads to a reward of $-1$ if selected by either agent and immediate episode termination. Thus, the last action corresponds to sabotaging the team's payoffs. By default, the game lasts for five timesteps.

We train ROTATE with both per-state and per-trajectory regret. To measure the extent to which the learned teammate policies engage in sabotage, we enumerate the 341 non-terminal states in the game and measure the probability of the sabotage action at each state for the last generated teammate policy. Fig. 4 shows that ROTATE with per-state regret has a near-zero probability of taking the sabotage action at all non-terminal states, while the per-trajectory regret objective leads to over a third of states that have P(S) near 1.0.

**RQ2b: Does per-state regret lead to improved generalization compared to per-trajectory regret? (Yes)** **RQ2a** demonstrated that ROTATE with per-state regret (ours) leads to teammate policies that sabotage less often in an illustrative matrix game, compared to ROTATE with per-trajectory regret. Here, we investigate whether the this translates to improved generalization against the unseen evaluation teammates. All configurations other than the teammate's policy objective are kept identical, including the data used to train the teammate value functions. Fig. 3b shows that ROTATE with per-state regret outperforms ROTATE with per-trajectory regret on all tasks except AA, confirming the superiority of per-state regret. As discussed in **RQ1**, we observe that AA is a layout where an ego agent is less susceptible to sabotage, due to the separated kitchen layout. App. C.4 presents additional experiments testing ROTATE with CoMeDi-style mixed-play rollouts, and alternative methods to compute per-state regret—ultimately finding that ROTATE outperforms all variations.

---

[2]Optimal behavior in AA still requires effective coordination due to layout asymmetry. In the "left" kitchen, the delivery zone is adjacent to the pots while the onions are farther, while in the "right" kitchen, the opposite is true. Thus, an optimal team consists of the "left" agent delivering finished soup, and the "right" agent placing onions in the pots—and indeed, we observe that teams of IPPO agents converge to this behavior.

**RQ3: Does the SXP return term of the ROTATE teammate generation objective improve learning and generalization? (Yes)** The ROTATE objective (Eq. 10) includes two terms: the per-state regret term and a SXP return term, introduced to ensure that the teammate collaborates well with its BR, even during SXP interactions. This helps ensure that the teammate is a good-faith collaborator. In Fig. 3b, the heldout return of an ablation of ROTATE where the SXP return term is removed is shown. We find that the ablated version performs signfiicantly worse than the non-ablated across $5/6$ of the tasks, confirming that the auxiliary return term plays and important role in the teammate generation objective.

**RQ4: Is the population buffer necessary for ROTATE to learn well? (Yes)** We hypothesize that collecting all previously generated teammates in a population buffer helps the ROTATE agent improve in robustness against all previously discovered conventions. On the other hand, if there is no population

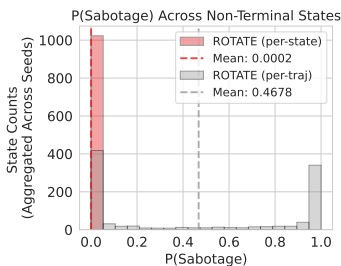

Figure 4: Probability of the sabotage action at all states in the sabotage game for ROTATE teammates trained with per-state regret (ours) vs per-trajectory regret. Results are aggregated across three trials.

buffer, then it becomes possible for the ROTATE ego agent to forget how to collaborate with teammate seen at earlier iterations of open-ended learning (Kirkpatrick et al., 2017), which creates the possibility that the ego agent and teammate generator oscillates between conventions. As shown in Fig. 6a, ROTATE without the population buffer attains lower evaluation returns than the full ROTATE method on all tasks except for AA, thus supporting the hypothesis that the population buffer improves ego agent learning. As discussed in **RQ1**, AA is a unique layout where agents can complete the task independently, even in the presence of an adversarial partner. As a corollary, there are few meaningful cooperative conventions that can be discovered, and no scenarios where convention mismatch leads to zero return (unlike LBF and FC).

## 8 DISCUSSION AND CONCLUSION

This paper reformulates AHT as an open-ended learning problem and introduces ROTATE, a regret-driven algorithm. ROTATE iteratively alternates between improving an AHT agent and generating challenging yet cooperative teammates by optimizing a per-state regret objective designed to discover teammates that exploit cooperative vulnerabilities while mitigating self-sabotage. Experiments on an illustrative matrix game demonstrate that the per-state regret objective mitigates self-sabotage. Extensive evaluations across six benchmark tasks demonstrate that ROTATE significantly enhances the generalization capabilities of AHT agents when faced with previously unseen teammates, outperforming baselines from both AHT and UED.

The current work has several limitations. First, while this paper provides intuitive justification and strong empirical evidence for the efficacy of the per-state regret objective, an exciting line of follow-up work is to formally define the concept of self-sabotage and theoretically analyze the properties of the proposed regret objectives. Second, the paper only validates ROTATE on two-agent, fully observable, and fully cooperative scenarios, which leaves the question of whether it scales to more complex scenarios for future work. Finally, this work has focused on the teammate generation phase of open-ended AHT. Future work might explore ego agent training methods that better handle the nonstationarity induced by open-ended teammate generation.

## REPRODUCIBILITY STATEMENT

As part of our submission, we provide additional information to ensure the reproducibility of our paper. Codes to run our experiments, including instructions to set up and run our experiments, are provided in this anonymous repository: https://anonymous.4open.science/r/rotate/. A detailed specification of the environments used in our experiments, including information on the reward function, length of the interaction episode, the environment's action space, and the environment's observation space, is provided in App. E. Meanwhile, a precise description of teammate policies from the holdout teammate policy sets used in evaluation, alongside the estimated performance of their respective best response policies, is provided in App. G. Finally, details of the hyperparameters and a specification of the compute infrastructure for our experiments are provided in App. F.

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
