# OpenReview forum: "ROTATE: Regret-driven Open-ended Training for Ad Hoc Teamwork"
_ICLR.cc/2026/Conference — Submitted to ICLR 2026_

### Official Review · Reviewer_1eRE · 2025-10-28

**Soundness:** 3
**Presentation:** 2
**Contribution:** 2
**Rating:** 4
**Confidence:** 4

**Summary:**

The paper reframes ad‑hoc teamwork as an open‑ended min–max process: a teammate generator maximizes the ego agent’s cooperative regret while the ego minimizes it, alternating over training. The key idea is state‑wise regret, optimized over SP/XP/SXP state distributions to discourage sabotaging teammates by construction. A population buffer stabilizes training under a non‑stationary teammate distribution. On six two‑agent tasks across LBF and Overcooked, the method outperforms strong baselines on 5/6 tasks, and a toy “destructive matrix game” illustrates that state‑wise regret strongly suppresses sabotage.

**Strengths:**

Clear problem reframing. Maximizing cross‑play with unknown partners is cast as minimizing cooperative regret, yielding a natural open‑ended training framework that aligns well with UED principles.

Objective‑level protection against sabotage. Jointly optimizing state‑wise regret on SP/XP/SXP exposes weaknesses while enforcing compatibility with a best‑response partner, which is more principled than adversarial diversity only from the initial state.

Reasonable empirical support. Diverse tasks, 9–13 “benign” evaluation partners, a normalized score (upper‑bounded by an estimated BR), and targeted ablations (trajectory vs. state‑wise regret; with/without population buffer).

**Weaknesses:**

1.Narrow experimental scope. Evidence is confined to two agents with full observability. The absence of results on larger multi‑agent and partially observable settings (e.g., SMAC, GRF)[1] weakens claims of scalability.

2.Lack of theory and stopping criteria. “Open‑endedness” lacks a formal definition and a practical stopping rule; there are no guarantees on coverage, convergence, or regret bounds, limiting deployability.

3.Comparative evaluation is incomplete. Empirical comparisons against LIPO[2], MACOP and rigorously budget‑matched versions of BRDiv/CoMeDi are missing; fairness requires equal interaction budgets/compute.

4.Positioning vs. prior work needs precision. The relationship to MACOP and related methods should be spelled out at the level of objective functions, “benign partner” constraints, stopping rules, and network design, to avoid conceptual conflation.

5.Evaluation and ablations need tightening (detail add‑ons).

The normalized metric depends on an estimated BR upper bound; report sensitivity to BR approximation error.

Provide weight sensitivity for SP/XP/SXP and justify the SXP term’s necessity.

Use more random seeds—open‑ended procedures can have high variance.

Clarify environmental assumptions (mid‑episode policy switching/state resets); give an approximation strategy when resets are unavailable.

Discuss computational complexity as the partner space expands, and what approximations to BR/regret are admissible with guarantees.

Ref:

[1]A survey of progress on cooperative multi-agent reinforcement learning in open environment

[2] Generating Diverse Cooperative Agents by Learning Incompatible Policies

**Questions:**

1.Scaling to many agents. How does the method avoid combinatorial blow‑up for 5–10 agents or more? Would centralized training with decentralized execution, hierarchical BR, population BR, or fictitious‑play‑style approximations be viable, and at what cost?

2.Multi‑modal partner distributions. If the partner distribution is genuinely multi‑modal, is a single ego policy sufficient? Would mixture‑of‑experts, latent‑variable policies, or distributionally robust objectives (e.g., CVaR) be required to capture distinct partner modes?

3.Formalizing open‑endedness. What is the precise criterion—coverage growth, novelty accumulation, or monotone regret reduction? Please provide operational metrics (coverage/novelty/regret) and a stopping rule, accompanied by evidence.

4.Human–AI collaboration. Can the method transfer to real human partners (e.g., Overcooked‑human)? Would demonstrations, preference modeling, or safety constraints be needed to bound the teammate generator’s search space?

5.Visualization and interpretability. Please include state‑level sabotage heatmaps, SXP vs. XP occupancy differences, teammate embedding visualizations, and term‑wise causal ablations to show where and why each component works.

6.Embodied multi‑agent and LLM integration[1]. Can the framework extend to embodied, partially observable, continuous‑control domains? Could LLMs serve as a teammate generator or a language‑mediated coordination channel for richer partner diversity and policy decomposition?

Ref:

[1]
Multi-agent embodied ai: Advances and future directions

---

> ### Author Response · Authors · 2025-11-18
>
> We thank the reviewer for recognizing the strengths of our clear problem reframing, the principled nature of our per-state regret teammate generation objective, and our strong empirical evaluation. We provide an initial response to the reviewer’s concerns below, asking for further clarification where appropriate, and plan to follow-up with additional results.
>
> ### **Weakness 1: Narrow Experimental Scope & Scaling to Multiple Agents**
>
> We must emphasize that we never made claims regarding the scalability of our method to the settings mentioned by Reviewer 1eRE. We have repeatedly mentioned across multiple sections (for example, the abstract and introduction) that our experiments are currently limited to the two-player fully observable settings. In Lines 479-481, we even highlighted that extensions to fully observable environments with more than two agents are left for future work. Finally, although extensions to these settings can improve the significance of the paper, note that many influential past teammate generation methods also restrict their experiments to two-player fully observable environments (Charakorn et al. 2023, Rahman et al. 2023, Sarkar et al. 2023). Further, extensions of existing two-player AHT methods to N-player environments is typically considered a paper-worthy contribution on its own (Wang et al. 2024, Charakorn et al. 2025).
>
> In contrast to producing a single teammate policy, generating teammates in environments with more than two players requires optimizing some objective defined over a collection of multiple joint policies of N -1 teammates. Then, there are various ways to train this collection of joint policies using MARL, depending on the number of agents and the available computing infrastructure. Although not scalable to a large number of agents, training centralized policies to control the N-1 agents can generate teammates that strongly coordinate their action selections at each time step. At the price of limiting the joint policy where teammates independently select their actions at each time step, CTDE MARL algorithms can also be repurposed to optimize the teammate generation objective. Finally, environments with many players may require techniques from mean-field multi-agent reinforcement learning. Ultimately, just as we repurposed the independent PPO algorithm to train a single teammate, many MARL techniques can be leveraged to generate joint teammates. See Yuan et al. 2023 for an overview of available techniques.
>
> ### **Weakness 2: Lack of Theory, Definition of Open-endedness, Stopping Criteria**
>
> Following the extensive work on open-ended learning methods to train RL agents, which dates back to 2020, we have clearly defined the concept of open-ended learning methods in Lines 130-132. This definition aligns with Hughes et al. (2024), who formulated open-ended methods as those that continuously generate novel and learnable tasks to improve an agent’s performance across a wide range of tasks. If we were given more space, we would definitely state this definition in more places in the main text. Please see our response to Question 3 for a more detailed discussion of how ROTATE satisfies their definition.
>
> Meanwhile, we strongly disagree with the need to formulate a stopping criterion. By definition (Hughes et al., 2024), open-ended learning methods are designed to run in perpetuity, continually generating tasks (or, in our case, teammate policies) that improve the trained agent’s capabilities. In practice, we only halt this continual learning process due to time and infrastructure constraints that limit our experiments. For similar reasons, none of the past open-ended learning papers that we cited in the related work section defined something resembling a stopping criterion for open-ended learning.
>
> However, we do provide empirical evidence of an *alternative* property that we might desire from an open-ended learning system: improvement against an unseen test set of teammates as training continues. In Appendix Fig. 8, we show the learning curves of all variants of ROTATE, constructed by evaluating ROTATE against the test set of teammates. Fig. 8 shows that ROTATE’s generalization returns increase as the open-ended learning iterations progress.

---

> ### Author Response · Authors · 2025-11-18
>
> ### **Weakness 3: Missing Comparison to LIPO/MACOP;  Budget-Matched Comparisons to BRDiv/CoMeDi**
>
> Given that we have compared our method to BRDiv (Rahman et al., 2023), we disagree with the need for further comparisons with LIPO (Charakorn et al., 2023). After all, LIPO is a specific case of BRDiv that assigns a particular weight for the cross-play minimization objective, as stated in the LIPO paper (see Appendix E of the LIPO paper). BRDiv and LIPO are contemporaneous methods, with BRDiv published at TMLR in May 2023, and LIPO being published at ICLR in May 2023. Given their equivalence, we argue that we’ve compared ROTATE with both LIPO and BRDiv.
>
> Meanwhile, MACOP’s (Yuan et al., 2023) use of evolutionary algorithms to generate teammates makes fair comparisons with ROTATE quite challenging. Specifically, evolutionary methods typically require collecting more experience than RL methods to achieve similar values of objective functions. Our closest approximation to MACOP is the PAIRED baseline, which retains most algorithmic details in the same way as MACOP, except for the teammate generation objective optimization that is performed using RL-based optimization. For more details, please see our answer to Weakness 4.
>
> Finally, our experiments *do* perform budget-matching between ROTATE and the baseline methods, as much as possible, as detailed in Appendix E.1. If provided with more space for the main text, we will make a statement that explicitly points to Appendix E.1 to indicate fair comparisons between methods.
>
> ### **Weakness 4: Relationship to MACOP**
>
> The difference between MACOP (Yuan et al., 2023) and ROTATE is twofold. First, unlike ROTATE, which maximizes cooperative regret at SP and XP states, MACOP generates teammates by maximizing cooperative regret only in the initial states during an interaction. Second, MACOP utilizes the cooperative regret as a fitness function for training teammates using an evolutionary approach. In contrast, ROTATE uses RL to maximize per-state regret during teammate generation.
> ### **Weakness 5a: Normalization with respect to Estimated BR Upper Bound**
>
> It is not possible to report sensitivity of the empirical returns to BR approximation error, as we do not know what the ground-truth BR returns are. Please let us know if the reviewer had something else in mind.
>
> ### **Weakness 5b: Weight Sensitivity for SP/XP/SXP**
>
> Can the reviewer clarify what they mean by weight sensitivity on XP and SP? To clarify, the 0.5 weighting in the subscript of Eq. 10 is simply notation to emphasize uniform sampling from all gathered SP and XP states (i.e. all available states encountered), and not a hyperparameter to be tuned. This caused another reviewer confusion as well, so we plan to fix this notation.
>
> We provide justification for the SXP term in Lines 297-298. In short, the SXP term is necessary in order to ensure that $\pi^{-i}$ is a good-faith collaborator even outside of SP states. To provide support, we will add an ablation study on the SXP term later in the discussion period.
>
> ### **Weakness 5c: More Random Seeds**
>
> Unfortunately, due to the extensive experiments and high computational cost of teammate generation experiments, we were only able to provide 3 seeds at submission time and may not be able to reproduce all experiments with additional seeds within the discussion period, given the other experiments asked for. If accepted, we will reproduce all results with 5 seeds.
>
> ### **Weakness 5d: Clarifying Environment Assumptions**
>
> ROTATE does not require any special environmental assumptions.
>
> Although our method is primarily explained with environment state resets, which reflects the actual code implementation, as we explain in Lines 300-313, a policy-switching strategy can be used instead if it’s not possible to reset the environment.
>
> Please note that the described policy switching strategy is *not* an assumption about the environment.  Since the original policy $\pi^1$  and the policy to switch to $\pi^2$ is known before the episode starts, and the random timestep t can be sampled before the episode starts,  we can always define a macro-policy that consists of the following logic:
>
> ```
> If t’ < t:
>     Deploy original policy pi1
> Else:
>     Deploy policy pi2
> ```
>
> This macro-policy can be executed in any RL environment without special assumptions.

---

> ### Author Response · Authors · 2025-11-18
>
> ### **Weakness 5e: Computational Complexity w.r.t. Number of Partners, Approximations to BR**
>
> Regarding **computational complexity**: we already provided analysis of the computational complexity of ROTATE and baselines with respect to the population size and number of objective updates in Appendix B.1.
>
> Regarding “approximations to BR/regret”: could the reviewer please clarify exactly what kind of guarantees they are looking for?
>
> In the current paper, we do not provide theoretical results. Instead, we provide extensive empirical evaluation of ROTATE, demonstrating that our learning-based approximations to the BR policy are sufficient to allow good generalization to unseen teammates. Informally, we experimentally found that it is important to train ROTATE’s teammate/BR to convergence at each iteration, indicating that the performance of ROTATE does depend on training high-quality teammates and BR’s.
>
> ### **Question 1: Scaling to Many Agents**
>
> Please see our answer for Weakness 1.
>
> We agree that techniques like centralized training with decentralized execution (CTDE) could be used to scale ROTATE to multiple agents, along with techniques described in Wang et al. (2023).
>
> We’re not sure what hierarchical BR, population BR, or fictitious-play approximations refers to. If the reviewer would like to continue discussing these techniques, could the reviewer please clarify?
>
> ### **Question 2: Multimodal Partner Distributions**
>
> Dealing with “multimodal” partner distributions is precisely the goal of training an ego agent, or AHT agent. There is an extensive literature on how to best design an ego agent to deal with varying partners, with most strategies revolving around agent modeling (Albrecht et al., 2018, Papoudakis et al., 2021, Wang et al., 2023). For all of these papers, the main idea is that the ego agent should learn to (1) model what *type* of a teammate it is interacting with in an episode, and (2) given the teammate’s type, deploy the *best-response* to that teammate type.
>
> The ROTATE ego agent does not use explicit agent modeling because it is orthogonal to what is considered within our paper. Please note that for fair comparison, no baselines in the paper use agent modeling either. Incorporating agent modeling would most likely improve the learning efficiency of the ego agents, and potentially lead to stronger generalization.
>
> However, although the ROTATE ego agent does not explicitly use agent modeling, it is possible that it has implicitly learned to model its teammates. Mon-Williams et al. (2025) performed a large scale analysis of thousands of RNN agent teams in AHT scenarios on Overcooked, finding that the agent modeling is an *emergent* feature of AHT training, even without explicit agent modeling losses or inductive biases. The finding also applies for ROTATE’s ego agent, which employs a recurrent S5 architecture.

---

> ### Author Response · Authors · 2025-11-18
>
> ### **Question 3: Formalizing Open-Endedness**
>
> **Defining Open-endedness**
>
> To formalize open-endedness, we can follow the definition of open-ended systems provided by Hughes et al. 2024, which we cite in Line 131: “From the perspective of an observer, a system is *open-ended* if and only if the sequence of artifacts it produces is both novel and learnable.” - Section 2.1, Hughes et al.
>
> Please refer to Hughes et al. directly for a full understanding of their definition of open-endedness. But to summarize briefly:
>
> Hughes et al. considers a system $S$ that produces a sequence of artifacts $X_t$, indexed by time $t$. An observer $O$ attempts to predict a new artifact $X_T$ based on its observations of past artifacts it has seen up until time $t$.
>
> Hughes et al. states that:
> >A system displays *novelty* if artifacts become increasingly unpredictable with respect to an observer’s model at any fixed time $t$, namely:
> >$$
> >\forall t, \forall T > t, \exists T’ > T: \mathbb{E}[\ell(t, T’)] > \mathbb{E}[\ell(t, T)].
> >$$
>
> This is measured by a *loss metric*, $\ell(t, T)$, where the first argument refers to the timestep that the observer has observed up until (i.e., it has observed artifacts $X_1, \cdots, X_t$, while the second argument refers to the artifact that the observer must now predict (i.e., $X_{T}$).
>
> While,
>
> >The system is *learnable* whenever conditioning on a longer history makes artifacts more predictable, namely:
> >$$
> >\forall T, \forall t < T, \forall T > t’> t: \mathbb{E}[\ell(t’, T)] < \mathbb{E} [\ell(t, T)].
> >$$
>
> **ROTATE displays both novelty and learnability according to the definition above**, if we consider the ego agent to be the observer, the sequence of generated teammates as the artifacts, and consider $\ell(t, T) $ to be the cooperative regret of the ego agent that has seen teammates generated up until iteration $t$, evaluated on the teammate generated at iteration $T$. We detail both arguments below.
>
> **Novelty**: at open-ended iteration $t+1$, ROTATE seeks to generate a teammate that *challenges* the ego agent from iteration $t$ by directly maximizing its cooperative regret. Thus, for T’ = T+1:
>
> $$
> \mathbb{E}[\ell(t, T’)] > \mathbb{E}[\ell(t, T)]
> $$
>
> **Learnability**: In the context of ROTATE, this condition states that for some *future unseen teammate from time T*, the cooperative regret of ROTATE’s ego agent from iteration $t’$ should be lower than the cooperative regret of ROTATE’s ego agent from an earlier iteration $t$.
>
> Intuitively, since all generated teammates are added to a population buffer that is uniformly sampled, ROTATE’s ego agent at future iterations should always be more capable than the ego agent at previous iterations.
>
> Empirically, we can see that this holds from the learning curve charts presented in Fig. 8 of the Appendix. In particular, the learning curves show that the normalized return against the test set of unseen teammates increases as the iterations increase. Crucially, the return is min-max normalized so that 1.0 represents the return achieved by a teammate with its best response, while 0.0 represents a known lower bound on returns. If the normalized return increases from iteration $t$ to $t’$, then the cooperative regret decreases in the same interval. See below for the formal argument:
>
> **Relationship between Normalized Return and Cooperative Regret**
>
> Let $\eta$ represent the lower return bound. For a single teammate $\pi^{-i}$, the definitions of cooperative regret (CR) and normalized returns are:
>
> Regret: $V(\pi^{-i}, BR) - V(\pi^{-i}, \pi^{ego})$
>
> Normalized Return: $(V(\pi^{-i}, \pi^{ego}) - \eta) / (V(\pi^{-i}, BR) - \eta)$
>
> First, let us transform regret to guarantee each term is greater than zero by adding and subtracting $\eta$:
>
> $(V(\pi^{-i}, BR) - \eta) - (V(\pi^{-i}, \pi^{ego}) - \eta)$
>
> Now, let $c$ represent the positively transformed return of the teammate with its best response, which is a constant as $\pi^{ego}$ varies:  $c := V(\pi^{-i}, BR) - \eta$.
>
> Further, let $R(t) = V( \pi^{-i}, \pi^{ego}_t) - \eta$  denote the positively transformed returns of the ego agent at iteration $t$.
>
> Thus, regret and normalized return of an ego agent at iteration $t$ can be rewritten:
>
> Regret: $Reg = c - R(t)$
>
> Normalized Return: $NRet = R(t) / c$
>
> Suppose that in the interval $[t, t’]$,  $R(t)$ increases monotonically with $t$. Let us now analyze the above as linear equations varying in $R(t)$. Since $R(t)>0$ and $c > 0$, normalized return (NRet) increases in the interval [t, t’], while regret (Reg) decreases in the same interval.
>
> **Providing a Stopping Rule**
>
> Please see our response to Weakness 2.

---

> ### Author Response · Authors · 2025-11-18
>
> ### **Question 4: Human AI Collaboration**
>
> Regarding generalization to **human partners**: in Appendix C.7, we evaluate ROTATE and CoMeDi, the best baseline, against the human-proxy teammates generated by behavior cloning on the standard Overcooked human gameplay dataset. We find that ROTATE achieves higher returns against the human proxy compared to CoMeDi across all five Overcooked layouts.
>
> Regarding the necessity of **bounding the teammate generator’s search space to achieve generalization to real humans**:
> The experiment described above indicates that in Overcooked, it is *not* necessary to bound the teammate generator’s search space to achieve generalization to humans. It’s possible that in more complex environments, such bounding is necessary. We are excited to explore how to perform open-ended teammate generation guided by human demonstrations, as a direction of future work. Such an approach could take inspiration from existing two-stage AHT methods such as GOAT, which provides a method to optimize cooperative regret over the latent space of a generative model trained on a human dataset (Chaudhury et al., 2025).
>
> ### **Question 5: Visualization and Interpretability**
>
> We address each reviewer request below:
>
>
> **State-level sabotage heatmaps**:
>
> To generate these, a necessary  assumption is that it’s possible to partition the set of states as “sabotage” states or “not sabotage” states, which is not the case in Overcooked or LBF. To study exactly this question, we design an iterated matrix game for studying sabotage, where there is a sabotage action available—see RQ2a and Figure 4.
>
> In this game, the state is defined as the history of joint actions. We evaluate the probability of taking the sabotage action at every state in the game. We find that ROTATE with our proposed per-state regret objective essentially never takes the sabotage action at any state. In contrast, ROTATE with per-trajectory regret (which is the objective used by other SP maximization-XP minimization papers) *does* take the sabotage action in a large count of states.
>
> **SXP vs XP Occupancy Differences**:
>
> Could the reviewer clarify what insight they are trying to gain here?
>
> XP states are from rollouts between the ego agent and the teammate, while SXP states are from rollouts between the teammate and the best response, starting from states encountered during XP. Our per-state regret objective attempts to encourage the teammate to behave similarly whether it is interacting with the ego agent or its best response.
>
> Is the reviewer asking us to experimentally confirm this via the occupancy visualization?
>
> **Teammate Embedding Visualizations**:
>
> We will post these results when available.
>
> **Termwise Causal Ablations**:
>
> For the ROTATE algorithm, term-wise ablations are often inappropriate, as many terms cannot be sensibly removed from the objective. However, we do provide many ablation-type experiments where we substitute the component used in ROTATE with an alternative component either considered by related work, or that we came up with ourselves.
>
> We describe the ablations that have been done already below, divided by whether they apply to the teammate generation phase or the ego agent training phase. For all ablations, the ablated version tends to perform worse than the version in ROTATE.
>
> - Teammate generation objectives:
>     - Figure 3b: ROTATE with per-trajectory regret is an ablation of the state distribution that cooperative regret is computed over (i.e., the subscript of the first expectation term in Eq 10)
>     - Figure 6a, red bars (Appendix): ROTATE with CoMeDi MP regret is an ablation where we substitute the per-state regret objective of ROTATE with the mixed play objective of CoMeDi
>     - Figure 6a, pink bars (Appendix): ROTATE with GAE regret (described in Appendix C.2)  is an ablation where we consider an alternative method to estimate ROTATE’s cooperative regret based on Generalized Advantage Estimators
> - Ego agent training
>     - Figure 6b, pink bars, (Appendix): ROTATE w/o population: this ablates the population pool maintained by ROTATE.
>
> We will perform an additional ablation of the 2nd term of Eq. 10, corresponding to return maximization over SXP states. Please let us know if there are further ablations we should consider.

---

> ### Author Response · Authors · 2025-11-18
>
> ### **Question 6: Embodied Multi-agent and LLM Integration**
>
> We break down the reviewer’s question by topic.
>
> **Extension to embodied scenarios**
>
> The ROTATE framework could be extended to embodied scenarios **if** a fast environment simulator is available. This requirement is shared by teammate generation algorithms for AHT in general, due to the high computational demands of teammate generation.
> For this reason, teammate generation algorithms (which ROTATE is similar to) are not tested on embodied AI domains.
>
> **Extension to partially observable scenarios**
>
> In principle, the ROTATE framework could be extended to partially observable scenarios. The only changes we foresee needing would be to use recurrent models for the teammate and the best response policies, to allow them to learn in the presence of partial observability.
>
> **Extension to continuous control scenarios**
>
> In principle, the extension of ROTATE to continuous control scenarios is straightforward, since we use PPO as a backbone RL algorithm (which was originally developed for continuous control).
>
> **Use of LLMs for teammate generation or language-based  coordination**
>
> This is out of the scope of our paper, but has been considered by recent papers in AHT (Li et al., 2025).
>
> ### **References**
>
> [1] Charakorn et al. Generating Diverse Cooperative Agents by Learning Incompatible Policies. ICLR 2023.
>
> [2] Rahman et al. Generating Teammates for Training Robust Ad Hoc
> Teamwork Agents via Best-Response Diversity. TMLR 2023.
>
> [3] Sarkar et al. Diverse Conventions for Human-AI Collaboration. NeurIPS 2023.
>
> [4] Wang et al. N-Agent Ad Hoc Teamwork. NeurIPS 2024.
>
> [5] Charakorn et al. n-LIPO: Framework for Diverse Cooperative Agent Generation Using Policy Compatibility. IEEE Transactions on Artificial Intelligence.
>
> [6] Yuan et al. A Survey of Progress on Cooperative Multi-agent Reinforcement Learning in Open Environment. arXiv 2023.
>
> [7] Hughes et al. Position: Open-Endedness is Essential for Artificial Superhuman Intelligence. ICML 2024.
>
> [8] Yuan et al. Learning to Coordinate with Anyone. DAI 2023.
>
> [9] Albrecht et al. Autonomous agents modelling other agents: A comprehensive survey and open problems. Artificial Intelligence, Vol. 258, 2018.
>
> [10] Agent Modelling under Partial Observability for Deep Reinforcement Learning. NeurIPS 2021.
>
> [11] Mon-Williams et al. Partner Modelling Emerges in Recurrent Agents (But Only When It Matters). arXiv 2025.
>
> [12] Chaudhary et al. Improving Human-AI Coordination through Online Adversarial Training and Generative Models. 2025.
>
> [13] Li et al. LLM-Assisted Semantically Diverse Teammate Generation for Efficient Multi-agent Coordination. ICML 2025.

---

### Official Review · Reviewer_EESm · 2025-10-31

**Soundness:** 2
**Presentation:** 2
**Contribution:** 2
**Rating:** 2
**Confidence:** 4

**Summary:**

The paper discuss ad hoc teamwork as an open-ended partner co-learning problem and introduces ROTATE, which optimizes a per-state cooperative regret objective while encouraging competent, non-adversarial teammates. The approach alternates between training the ego policy and generating partner policies using state distributions from self-play, cross-play, and switched-play interactions, aided by a population buffer. Experiments across cooperative benchmarks report improved generalization to unseen partners.

**Strengths:**

The paper foregrounds the self-sabotage failure mode in open-ended partner generation, clearly articulating why partners that deliberately depress cross-play (XP) can inflate training signals yet harm zero-shot coordination; this diagnosis sharpens evaluation design (e.g., beyond average XP) and motivates principled mitigation objectives.

**Weaknesses:**

- Eq. 10 employs a fixed 0.5/0.5 weighting with no analytical justification, and the experiments do not analyze this hyperparameter.

- The method section has poor readability, with unclear logic and difficult-to-follow exposition.

- Missing ZSC-side baselines, especially some open-ended methods like COLE [1] and E3T [2].

**Questions:**

- Relationship to [3] and [4]: The method currently appears very similar to these two paper—could the authors clarify whether, under certain conditions, it degenerates to XP-min? How do the weights in Eq. 10 relate to XP-min’s hyperparameter ?

- Role of regret: The paper lacks analysis of regret’s actual effect. Do the generated partners indeed exhibit the property of “coordinating with a BR while exposing the ego’s weaknesses without engaging in self-sabotage”? Does Eq. 10 admit any theoretically provable guarantee that suppresses self-sabotage?

Reference

[1] Li, Yang, et al. "Cooperative open-ended learning framework for zero-shot coordination." International Conference on Machine Learning. PMLR, 2023.

[2] Yan, Xue, et al. "An efficient end-to-end training approach for zero-shot human-AI coordination." Advances in neural information processing systems 36 (2023): 2636-2658.

[3] Charakorn, Rujikorn, Poramate Manoonpong, and Nat Dilokthanakul. "Diversity is not all you need: Training a robust cooperative agent needs specialist partners." Advances in Neural Information Processing Systems 37 (2024): 56401-56423.

[4]Sarkar, Bidipta, Andy Shih, and Dorsa Sadigh. "Diverse conventions for human-AI collaboration." Advances in neural information processing systems 36 (2023): 23115-23139.

---

> ### Author Response · Authors · 2025-11-20
>
> We thank the reviewer for their valuable feedback. We provide an initial response to the reviewer’s concerns below, asking for further clarification where appropriate, and plan to follow-up with additional results.
>
> ### **Weakness 1: 0.5 Weighting in Eq 10**
>
> This weighting is simply notation to emphasize uniform sampling from all gathered SP and XP states (i.e. all available states encountered), and not a hyperparameter to be tuned. While it’s possible to imagine sampling from the gathered states using an alternative distribution, we see no compelling reason to do so.
>
> We apologize for the misunderstanding and will fix the notation to explicitly refer to uniform sampling.
>
> ### **Weakness 2: Readability of Methods Section**
>
> We are sorry that the reviewer found the logic and exposition of the methods section of the paper lacking and would very much like to improve it. However, we need some clarification on exactly what the reviewer found confusing. For example, which particular subsections out of Sections 4, 5, or 6? These sections present ROTATE in full detail, and are all absolutely crucial towards understanding our method.
>
> ### **Weakness 3: Missing COLE and E3T as Baselines**
>
> **E3T as a Baseline**
>
> E3T (Yan et al., 2023) is not an open-ended method, although it does not fit into the typical 2-stage framework (as we mention at Lines 115-116).
>
> The main idea of E3T’s teammate generation is to train the ego agent with a randomly perturbed copy of itself. More precisely, the partner policy is $\pi_p = \epsilon \pi_r + (1 - \epsilon) \pi_e$ (Eq 4 of E3T paper), where $\pi_r$ is the random policy, and $\pi_ego$ is the ego agent policy. In other words, it is randomly perturbed self-play.
>
> Of course, we include other non-open-ended AHT baselines in our comparison to ROTATE, so that by itself is not a reason to exclude E3T from comparison. However, the independent evaluation of E3T performed by the ZSC-eval benchmark (Wang et al. 2024) shows that E3T (a non-population-based method) generally performs similarly to FCP with the lowest population sizes of 12 and 24 on Overcooked (Figures 4-6 of ZSC-eval paper), and worse than FCP in Google Research Football’s “3 vs 1 with Keeper” scenario (Table 3 of ZSC-eval paper). Please note that we run FCP with a population size of 110 agents (see paragraph 2, Appendix E.1), which is similar to the population size of 96 agents employed by the original FCP paper (Strouse et al. 2021).
>
> For fair comparison, we would also need to ablate out the partner modeling module of E3T’s ego agent, since our paper does not consider explicit partner modeling, and no other method in the paper uses explicit partner modeling.
>
> Given the above reasons, we find it unlikely that E3T would outperform ROTATE and do not think it’s an interesting baseline.
>
> **COLE as a Baseline**
>
> We omitted COLE by mistake from the related work and will add a citation and discussion. There are several differences between COLE and ROTATE, but the most important is that ROTATE directly pursues the objective of AHT, which is generalization to unseen partners (see  Sections 4 and 5), while COLE does not directly pursue that objective, but rather optimizes for the cooperative diversity heuristic of “cooperative incompatibility”, as many other teammate generation methods do.
>
> Similar to E3T, according to the ZSC-eval benchmark paper (Wang et al. 2024), COLE performs similarly to or worse than FCP in the 7/9 of Overcooked tasks considered (Figures 4-5 of ZSC-eval paper) for both moderate and expert evaluation partners (Figure 6 of ZSC-eval paper). It also performs worse than FCP in the Google Research Football task (Table 3 of ZSC-eval paper). Thus, similar to E3T, we find it unlikely to outperform ROTATE.

---

> ### Author Response · Authors · 2025-11-20
>
> ### **Question 1: Relationship to CoMeDi, SpecTRL, and XP-minimization**
>
> We specifically compare and contrast ROTATE against CoMeDi (Sarkar et al. 2023) and SpecTRL (Charakorn et al. 2024) in the sections below. For a discussion of the general relationship of ROTATE to existing teammate generation methods optimizing SP and XP losses (i.e., XP-min techniques), please see our response to Reviewer NkUx under, “Weakness 2b: Novelty Over Existing SP and XP-based Objectives”.
>
> **Relationship to CoMeDi**:
>
> There are several major differences between CoMeDi and ROTATE.
>
> First, CoMeDi is a *two-stage method*, while ours is an open-ended method. While CoMeDi proposes generating the teammate population in an interactive way, it is still necessary to pre-specify the desired number of teammates, and train an ego agent on the outputted partner population. This is clearly stated in Algorithm 2, Appendix A.2 of the CoMeDi paper, which provides the simplified pseudocode of CoMeDi. Note that Algorithm 2 takes the number of policies to generate as an input, and outputs the population of teammates.
>
>  In other words, CoMeDi focuses on how to generate a diverse set of teammates, but it is still necessary to train an ego agent to best-respond to the set of generated teammates. In contrast, ROTATE interleaves teammate generation and ego agent training, focusing on generating teammates that challenge the ego agent’s cooperative ability at each iteration, effectively training the ego agent against a continuous stream of novel teammates.
>
> Second, while CoMeDi’s proposed teammate generation objective bears some similarity to the per-state regret teammate generation objective of ROTATE, CoMeDi assesses SP, XP, and mixed-play (MP) with respect to the generated set of teammates. Thus, their objective *does not have a clear connection* to the true objective of AHT: generalization of the ego agent’s policy to an unknown test set of teammates (see Section 5). In contrast, ROTATE’s overall objective (Eq. 7, minimax cooperative regret) is directly designed to minimize the worst-case regret of the ego agent, for any teammate policy, which is a reasonable AHT objective when we assume lack of knowledge of teammates seen in eval.
>
> Finally, please note that we already discuss the relationship between CoMeDi and ROTATE extensively.
>
> - We explicitly compare against CoMeDi as a baseline in the main paper’s experiments.
> - We include an experiment where we directly compare ROTATE’s per-state regret objective to CoMeDi’s mixed play objective *within our open-ended learning framework* in Appendix Fig 6a (red bars, ROTATE+CoMeDi MP).
> - In Appendix C.1, we provide a detailed discussion of how mixed play (MP) relates to ROTATE’s per-state regret teammate generation objective. A pointer is currently provided in the description of CoMeDi (Appendix B, Line 977), but we will improve clarity by adding a pointer in the main text as well.
> - In Appendix B.1, we analyze the computational complexity of ROTATE relative to all baselines, including CoMeDi (Lines 997-1001).
>
> Given the additional page allowed for the discussion and camera ready phase, we can move some of these analyses to the main text, if the reviewer agrees it would be useful.

---

> ### Author Response · Authors · 2025-11-20
>
> **Relationship to SpecTRL**:
>
> There are several major differences between SpecTRL and ROTATE. We are happy to add the below discussion to the Appendix.
>
> Like CoMeDi, the first and most crucial difference is that SpecTRL is a two-stage AHT method. The key idea in SpecTRL is to generate a population of initial partners using XP-minimization (XP-min Partners), which is known to result in self-sabotaging teammates. To fix this, SpecTRL proposes to distill a *new* set of non-self-sabotaging teammates from the self-sabotaging teammate population by training a new set of teammates to coordinate with the XP-min Partners — these are called the Distilled Partners.
>
> As shown in Fig 5 in the SpecTRL paper, the SpecTRL pipeline is:
>
>
> ```
> XP-min Training -> XP-min Partners -> Specialization Transfer -> Distilled Partners -> Generalist Training -> Generalist.
> ```
>
> The SpecTRL paper refers to the ego agent, or the AHT agent, as the *Generalist*. Thus, clearly, the training process of SpecTRL separates the teammate generation stage from the AHT agent training stage, with no interaction between the stages.
>
> Regarding the reviewer’s question on **whether ROTATE’s per-state regret objective reduces to XP-min**: The answer is that it **never** reduces to XP-min under any hyperparameter combination. Our paper refers to XP-min as *per-trajectory regret*. Please see Figure 2 in the paper for a visualization of per-trajectory regret compared to per-state regret. To transform ROTATE’s per-state regret objective into per-trajectory regret, one would need to remove the maximization of regret over SP and XP states (i.e. the dotted lines in Fig 2, right), and the maximization of best-response returns from SP states (i.e. the 2nd term in Eq 10).
>
> Regarding the reviewer’s question on **how the weights in Eq. 10 relates to XP-min’s hyperparameter**: We hope our answer to Weakness 1 clarifies this point. But to be explicit, the weights in Eq. 10 are not hyperparameters, and there is no relationship to the $\lambda$ hyperparameter trading off between SP maximization and XP minimization in the XP min objective (Eq 4, SpecTRL paper (Charakorn et al., 2024)).
>
>
> ### **Question 2: Role of Regret & Theoretical Guarantees on  Self-Sabotage**
>
> We break down the reviewer’s question on the actual effect of regret in ROTATE into several parts.
>
> - Do the generated partners exhibit the property of coordinating with a BR?
>     - Following the notation in our paper, let $\pi^{-i}$ denote the teammate. Throughout the paper, BR refers to $BR(\pi^{-i})$. By the definition of best response, the BR is trained to coordinate with the teammate.
>     - Further, the 2nd term in Eq 10 explicitly encourages the BR to coordinate with the teammate, even outside of SP states.
>
> - Do the generated partners expose the ego agent’s weaknesses?
>     -  This is a direct consequence of optimizing regret. High regret means the ego agent's returns against a teammate from starting state $s$ is far from optimal. In that sense, maximizing regret will always expose the ego agent's cooperative weakness.
>
> - Do the generated partners avoid engaging in self-sabotage?
>     - We *do* perform an experiment analyzing the effect of ROTATE’s per-state regret compared to per-trajectory regret via a repeated matrix game specially designed to study self-sabotage (Fig 4 in Section 7). In short, we find that per-state regret decreases the count of states where the teammate takes the “sabotage” action to near-zero. Please see RQ2a for the full text explaining the experiment.
>     - We also analyze whether the per-state regret objective leads to improved generalization in Fig 3b. Please see Section 7, RQ2b for this experiment.

---

> ### Author Response · Authors · 2025-11-20
>
> Regarding the reviewer’s question on **theoretical provable guarantees of suppressing self-sabotage**:
>
> First, as this has been a point of confusion for other reviewers, we’d like to clearly state that the main point of this paper is not solving self-sabotage, nor do we claim to do so. The main contributions of our paper are (1) fundamentally reformulating the AHT problem as a regret-based, open-ended learning problem, which allows directly optimizing for the generalization against unseen teammates, and (2) providing an initial empirical algorithm to instantiate the open-ended learning problem. As such, we only mention that our per-state objective *mitigates* self-sabotage (see Line 275, 472).
>
> Indeed, our paper does not provide theoretical guarantees of suppressing self-sabotage. We explicitly acknowledge this limitation and propose it as a line of future work at Lines 476-479. As the reviewer is clearly aware, self-sabotage has been a major challenge for SP and XP based teammate generation methods, and has motivated several papers, including the CoMeDi and SpecTRL papers discussed above. We agree that if we were claiming to solve self-sabotage, then we would need to (1) carefully define self-sabotage, (2) perform a thorough theoretical and empirical analysis of how a proposed solution reduces self-sabotage.
>
> Instead, our paper provides experiments confirming that:
> - The per-state regret objective *mitigates* self-sabotage compared to the per-trajectory regret objective (i.e. XP-min)
> - The per-state regret is helpful towards the generalization performance of ROTATE compared to per-trajectory regret (i.e. XP-min)
>
> ### **References**
>
> [1] Yan et al. An Efficient End-to-End Training Approach for Zero-Shot Human-AI Coordination. NeurIPS 2023.
>
> [2] Wang et al. ZSC-Eval: An Evaluation Toolkit and Benchmark for Multi-agent Zero-shot Coordination. NeurIPS 2024.
>
> [3] Strouse et al. Collaborating with Humans without Human Data. NeurIPS 2021.
>
> [4] Charakorn et al. Diversity Is Not All You Need: Training A Robust Cooperative Agent Needs Specialist Partners. NeurIPS 2024.

---

### Official Review · Reviewer_NkUx · 2025-11-02

**Soundness:** 2
**Presentation:** 1
**Contribution:** 2
**Rating:** 2
**Confidence:** 4

**Summary:**

The paper proposes ROTATE, a regret-driven open-ended training framework for ad hoc teamwork that reframes zero-shot coordination as minimizing worst-case cooperative regret, i.e., $\min_{\pi^{ego}}\max_{\pi^{-i}}\mathbb{E}[\mathrm{CR}(\pi^{ego},\pi^{-i})]$. It introduces a per-state regret objective coupled with an SXP term that maximizes payoff with a best-response partner to discourage sabotage, and alternates teammate generation with ego learning using a population buffer. On Overcooked and Level-Based Foraging, ROTATE outperforms UED and teammate-diversification baselines with unseen partners, and ablations attribute gains to the per-state objective and the buffer.

**Strengths:**

- Comprehensive treatment of ZSC/ad hoc teamwork: the paper unifies teammate generation and ego learning via a cooperative-regret min–max objective $ \min_{\pi^{ego}}\max_{\pi^{-i}}\mathbb{E}[\mathrm{CR}]$, making assumptions and evaluation protocol explicit.

**Weaknesses:**

- Clarity and exposition: the paper is difficult to follow; the core algorithmic loop (who updates when, how SP/XP/SXP are sampled/weighted, and how the BR is trained/used) is buried under notation, so the end-to-end procedure remains unclear even after multiple readings.
- Mischaracterization of the gap: (a) the claim that most ZSC/AHT methods are two-stage is outdated—recent open-ended or end-to-end approaches already move beyond fixed teammate sets (e.g., COLE [1], E3T [2], TrajeDi [3]); (b) the comparison to current work is incomplete, especially where prior methods already leverage SP/XP and mixed-play/SXP-style rollouts (e.g., CoMeDi [4]), making the incremental novelty of the proposed per-state regret $J_{\text{state}}$ hard to isolate.
- Anti-sabotage rationale under-specified: the paper asserts that coupling per-state regret with an SXP best-response term mitigates sabotage, but offers little intuitive or theoretical support (e.g., no conditions under which maximizing SXP payoff with BR implies low sabotage against arbitrary partners, no analysis of bias induced by approximate BR or sampling); more formal justification or counterexample analysis is needed.

References

[1] Li, Y., Zhang, S., Sun, J., Du, Y., Wen, Y., Wang, X., and Pan, W. 2023. Cooperative Open-ended Learning Framework for Zero-Shot Coordination. In Proceedings of the 40th International Conference on Machine Learning (ICML 2023). Proceedings of Machine Learning Research, 202:20470–20484.

[2] Yan, X., Guo, J., Lou, X., Wang, J., Zhang, H., and Du, Y. 2023. An Efficient End-to-End Training Approach for Zero-Shot Human-AI Coordination. In Proceedings of the Thirty-Seventh Conference on Neural Information Processing Systems (NeurIPS 2023).

[3] Lupu, A., Cui, B., Hu, H., and Foerster, J. 2021. Trajectory Diversity for Zero-Shot Coordination. In Proceedings of the 38th International Conference on Machine Learning (ICML 2021). Proceedings of Machine Learning Research, 139:7204–7213.

[4] Sarkar, B., Shih, A., and Sadigh, D. 2023. Diverse Conventions for Human-AI Collaboration. In Proceedings of the Thirty-Seventh Conference on Neural Information Processing Systems (NeurIPS 2023).

**Questions:**

- Does the combination of per-state regret on SP/XP and the SXP best-response payoff formally or intuitively guarantee reduced sabotage (i.e., lower probability of destructive actions), and under what assumptions on BR optimality and sampling?
- Can an agent maximize $J_{\text{state}}$ on SP/XP while keeping high SXP payoff yet still sabotage arbitrary non-BR partners (e.g., collusion with BR)? Is there any bound linking SXP payoff to sabotage rate against unseen partners?
- How sensitive is the anti-sabotage effect to the weighting between SP/XP and SXP and to environments without reliable state resets/cut-ins? Please provide analysis or ablations.

---

> ### Author Response · Authors · 2025-11-20
>
> We thank the reviewer for taking the time to read our paper and provide feedback. We provide an initial response to the reviewer’s concerns below, asking for further clarification where appropriate, and plan to follow-up with additional results.
>
> ### **Weakness 1: Clarity & Exposition**
>
> We are sorry that the reviewer found it difficult to follow the paper and would very much like to improve this aspect. Other than the comments about the core algorithmic loop and the mathematical notation, could the reviewer clarify what else they found confusing? For example, were there any particular subsections out of Sections 4, 5, or 6 that are unclear? Taken together, these sections present ROTATE in full detail.
>
> Regarding the specific feedback from the reviewer:
>
> - *Core algorithmic loop*: Due to lack of space at paper submission time, this was placed in the Appendix, and a pointer is provided at the beginning of Section 6. The core algorithmic loop is presented as Alg 1, with the TeammateGenerator and EgoUpdate functions presented in Alg 2 and Alg 3. Accompanying text describes the algorithms. This provides “who updates when, how SP/XP/SXP are sampled/weighted, and how the BR is trained/used”. Given the additional page provided by ICLR, we can move the core loop (Alg 1) to the main text to improve clarity.
>
> - *Mathematical Notation*: We introduced some deal of mathematical notation because we wanted to err on the side of precision while being concise. Could the reviewer point out which mathematical notations they found confusing?
>
> ### **Weakness 2a: Mischaracterization of the Gap**
>
> We disagree that we have mischaracterized the current state of AHT, by stating that *most* methods fall in the two-stage framework. We do not claim that *all* AHT methods fall in the two-stage framework and explicitly acknowledge that there are a few recent papers that do not follow the two-stage framework at Line 113-116 in our paper, and explain how ROTATE is novel compared to them. Further, prior to 2022, the vast majority of AHT papers collected by Mirsky et al. (2022) consider the two-stage framework.
>
> Regarding COLE and E3T: we specifically discussed E3T at Lines 113-116. We omitted COLE by mistake, but will include a citation in the next version. There are several differences between COLE and ROTATE, but the most important is that ROTATE directly pursues the objective of AHT, which is generalization to unseen partners (see  Sections 4 and 5), while COLE does not directly pursue that objective, but rather optimizes for the cooperative diversity heuristic of “cooperative incompatibility”, as many other teammate generation methods do.
>
> We also disagree that AHT has moved *beyond* the two-stage framework. Papers published within the last year have specifically considered the two-stage framework (Villin et al., 2025; Chaudhary et al., 2025; Charakorn et al., 2025a, Erlebach et al., 2024, Wang et al., 2024).
>
> Among the papers that the reviewer cited, please note that CoMeDi and TrajedDi are actually two-stage methods, since they specifically address the problem of generating a set of diverse teammates for an ego agent to train against (i.e., stage 1 in our two-stage framework). After the teammates are generated, an ego agent must be trained against the set of generated teammates.

---

> ### Author Response · Authors · 2025-11-20
>
> ### **Weakness 2b: Novelty Over Existing SP and XP-based Objectives**
>
> Indeed, self-play (SP) and cross-play (XP) based objectives have been considered in prior work, which is why we rely on this terminology to explain ROTATE’s per-state regret objective in Section 6. We expected that this would position ROTATE with respect to prior work, rather than obscure ROTATE’s novelty.
>
> There are several crucial differences between existing SP and XP objectives and ROTATE’s objective.
>
> First, in ROTATE, SP and XP is computed with respect to the *ego agent*, which is motivated by our novel reformulation of past AHT training objectives when the set of teammates encountered during evaluation is unknown (mentioned as Contribution 1 in Line 97, and  detailed in Section 5). In contrast, in existing AHT papers that have considered SP and XP objectives (for example, LIPO (Charakorn et al, 2025b), BRDiv (Rahman et al., 2023), CoMeDi (Sarkar et al., 2023), and LBRDiv (Rahman et al., 2024)), SP and XP are both computed with respect to the *population of generated teammates*.
>
> Despite its subtlety, this difference is crucial because maximizing the ego agent policy’s regret allows us to connect our version of SP and XP optimization (i.e., cooperative regret) to the true objective of AHT: the ego agent’s out-of-distribution generalization to an unknown test set of teammates (Section 5). Past works’ disregard of the ego agent policy during teammate generation, means that generated teammates may not help the ego agent learn to improve its generalization capabilities to unknown teammates. This misalignment between the teammate generation and ego agent evaluation objective results in these methods being outperformed by ROTATE in our experiments.
>
> Second, even beyond this difference, the existing SP and XP objectives are *per-trajectory regret* objectives, while ROTATE’s objective is a *per-state regret* objective. This is a substantial difference, especially since per-trajectory regret leads to self-sabotage, which has been observed by prior work (Cui et al., 2023; Sarkar et al., 2023). Our experiments demonstrate that:
>
> - ROTATE has improved generalization compared to an ablation of ROTATE with per-trajectory regret (Fig 3b)
>
> -  Per-state regret mitigates sabotage compared to per-trajectory regret in a matrix game designed to study self-sabotage (RQ2a, Fig 4).
>
> If the reviewer finds this discussion useful, we are happy to incorporate this additional contextualization to the manuscript.
>
> ### **Weakness 2c: Novelty over CoMeDi and Mixed Play**
>
> Please note that CoMeDi is explicitly included as a baseline in our experiments. The conceptual difference between our proposed per-state regret and mixed play is described in Appendix C.1. A pointer is currently provided in the description of CoMeDi (App. B, Line 977), but we will improve clarity by adding a pointer in the main text as well.
>
> Further, in addition to including CoMeDi as a baseline in our experiments, we include an experiment where we directly compare ROTATE’s per-state regret objective to CoMeDi’s mixed play objective *within our open-ended learning framework* in App. Fig 6a (red bars, ROTATE+CoMeDi MP).

---

> ### Author Response · Authors · 2025-11-20
>
> ### **Weakness 3: Anti-Sabotage Rationale Underspecified**
>
> We provide an intuitive justification for why optimizing per-state regret on SP and XP states reduces sabotage from Lines 293-297.
>
> We also perform experiments analyzing to what degree the per-state regret objective mitigates self-sabotage compared to per-trajectory regret via a repeated matrix game specially designed to study self-sabotage (Fig 4). We also analyze whether the per-state regret objective leads to improved generalization in Fig 3b. Please see Section 7, RQ2a and RQ2b for the text explaining these experiments.
>
> Please note that we do not claim to solve the self-sabotage problem, (see the abstract, and Lines 81-101)--to put it bluntly, it is not the point of this paper. The main contributions of our paper are (1) fundamentally reformulating the AHT problem as a regret-based, open-ended learning problem, which allows directly optimizing for the generalization against unseen teammates, and (2) providing an initial empirical algorithm to instantiate the open-ended learning problem.
>
> Sabotage has been a major challenge for SP and XP based teammate generation methods and has motivated several papers (Cui et al., 2023; Sarkar et al., 2023, Charakorn et al. 2024c). We agree that if we were claiming to solve self-sabotage, then we would need to (1) carefully define self-sabotage, (2) perform a thorough theoretical and empirical analysis of how a proposed solution reduces self-sabotage. We explicitly acknowledge this limitation and line of future work at Lines 476-479.
>
> The only reason why we even mention self-sabotage in this paper is that we found it necessary to mitigate it in order for ROTATE to work.
> As such, we only mention that our per-state objective *mitigates* self-sabotage (see Line 275, 472). Since it is an important part of ROTATE, we do perform experiments confirming that (1) the per-state regret objective *mitigates* self-sabotage, and (2) is helpful towards the generalization performance of ROTATE.
>
> Please let us know if this discussion addresses the reviewer’s concern. If so, we can add a discussion to the paper that clarifies the limited extent of our contribution to the general sabotage problem.
>
> ### **Question 1: Does the combination of per-state regret on SP/XP and the SXP best-response payoff formally or intuitively guarantee reduced sabotage (i.e., lower probability of destructive actions), and under what assumptions on BR optimality and sampling?**
>
> For the part of the question on self-sabotage, please see above.
>
> We hope to add an ablation experiment where we remove the SXP best-response payoff from ROTATE’s teammate generation objective before the discussion phase ends. We expect that the experiment will show that ROTATE has slightly worse generalization performance without the auxiliary SXP best-response payoff, but will not suffer major effects, as the per-state regret part is the core part of the objective.
>
> For the part of the question on BR optimality, since our paper does not make theoretical claims, we do not have formal assumptions on BR optimality. Informally, we experimentally found that it is important to train ROTATE’s teammate/BR to convergence at each iteration, indicating that the performance of ROTATE does depend on training high-quality teammates and BRs.
>
> For the part of the question on BR sampling, could the reviewer clarify what this means?
>
> ### **Question 2: Can an agent maximize on SP/XP while keeping high SXP payoff yet still sabotage arbitrary non-BR partners (e.g., collusion with BR)? Is there any bound linking SXP payoff to sabotage rate against unseen partners?**
>
> Could the reviewer clarify the first question? We are confused about why sabotage against unseen partners is relevant. After all, the role of generated teammate policies in our work is *limited to partnering the ego agent policy during training*. Thus, mitigating self-sabotage against the current ego agent policy is sufficient for providing novel and learnable teammates that also highlight the cooperative weaknesses of the ego agent, which we argue is crucial for improving the generalization capability of the ego agent during training (i.e., see Section 5). Although self-sabotage against unseen partners may be relevant if the generated teammate policies are used during evaluation, we never use them for this purpose and deem self-sabotage against unseen partners as irrelevant.
>
> Our per-state regret objective was designed to prevent the teammate from sabotaging the *ego agent* only, relative to its performance with the best-response agent, so that its behavior is useful for training the ego agent. We do not claim that the teammate would avoid sabotaging any possible teammate. We will add statements that clarify this point in the updated version of the paper.

---

> ### Author Response · Authors · 2025-11-20
>
> ### **Question 3: How sensitive is the anti-sabotage effect to the weighting between SP/XP and SXP and to environments without reliable state resets/cut-ins? Please provide analysis or ablations.**
>
> We will add experiments analyzing the impact of the weighting of terms in per-state regret on sabotage using our sabotage matrix game. We will post the results when ready.
>
> Regarding environments without reliable state resets/cut-ins:
> As described in lines 301-313, if it’s not possible to reset the environment or the resetting is unreliable, we can still execute ROTATE using *policy-switching*, an idea introduced by Sarkar et al. 2023. Policy switching does not rely on environment resetting.
>
> Please note that the policy-switching strategy is *not* an assumption about the environment.  Since the original policy $\pi^1$  and the policy to switch to $\pi^2$ is known before the episode starts, and the random timestep t can be sampled before the episode starts,  we can always define a macro-policy that consists of the following logic:
>
> ```
> If t’ < t:
>     Deploy original policy pi1
> Else:
>     Deploy policy pi2
> ```
>
> ### **References:**
>
> [1] Mirsky et al. A Survey of Ad Hoc Teamwork Research. EUMAS 2022.
>
> [2] Villin et al. A Minimax Approach to Ad Hoc Teamwork. AAMAS 2025.
>
> [3] Chaudhary et al. Improving Human-AI Coordination through Online Adversarial Training and Generative Models. 2025.
>
> [4] Charakorn et al. n-LIPO: Framework for Diverse Cooperative Agent Generation Using Policy Compatibility. IEEE Transactions on Artificial Intelligence, 2025.
>
> [5] Erlebach et al. RACCOON: Regret-based Adaptive Curricula for Cooperation. 2024.
>
> [6] Wang et al. N-Agent Ad Hoc Teamwork. NeurIPS 2024.
>
>
> [7] Charakorn et al. Generating Diverse Cooperative Agents by Learning Incompatible Policies. ICLR 2023.
>
> [8] Rahman et al. Generating Teammates for Training Robust Ad Hoc
> Teamwork Agents via Best-Response Diversity. TMLR.
>
> [9] Sarkar et al. Diverse Conventions for Human-AI Collaboration. NeurIPS 2023.
>
> [10] Rahman et al. Minimum coverage sets for training robust ad hoc teamwork agents. AAAI 2024.
>
> [11] Cui et al. Adversarial Diversity in Hanabi. ICLR 2023.
>
> [12] Charakorn et al. Diversity is Not All You Need: Training A Robust Cooperative Agent Needs Specialist Partners. NeurIPS 2024.

---

> > ### Comment · Reviewer_NkUx · 2025-11-27
> >
> > The authors' rebuttal addresses some of my concerns, and I have decided to maintain the original score.

---

### Official Review · Reviewer_xw5y · 2025-11-03

**Soundness:** 3
**Presentation:** 2
**Contribution:** 2
**Rating:** 4
**Confidence:** 5

**Summary:**

This tackles the problem of Ad-Hoc Teamwork through iterative training diverse teams.

The diffuclty in AHT is that your AHT agent might be robust with respect to some, apparently diverse population of teammates, but not others. The paper claims that this is due to a small traning set. Their idea is to use iterative training in order to minimise co-operative regret.

The iterative idea is obviously not new and the related work contains some algorithms which maintain a diverse set of opponents. ROTATE uses a minimax approach, similarly to e.g. Villein et al, but finds a worst-case policy,rather than a distribution, at each step. This makes me think that the authors have not looked at the related work in sufficient detail.

I would have liked the algorithm to be more precisely defined in the main paper: Eq. 7 says that they use a minimax approach, but $\Pi^{-i}$ is not defined until Section 6, and only really discussed in detail in the appendix. Since many other works use iterative training, I suppose that the real open-endedness is the generation of new partners, rather than the iterative nature of the training.

**Strengths:**

+ Interesting notion of regret
+ Good comparison with related work.

**Weaknesses:**

- The authors could have done a better job of identifying which component is more important: the notion of regret, the way the teammates are generated, etc.
- Unclear novelty.
- Lack of clarity and theoretical discussion.

**Questions:**

Can you explain exactly how you used the baselines? From my reading of the appendix, it seems that you only took some aspect of these approaches, and adapted them to your framework, rather than have done a direct comparison.

---

> ### Author Response · Authors · 2025-11-18
>
> We thank Reviewer xw5y for recognizing the novelty of our proposed regret objective, and the strength of our comparison with related work.
>
> We provide an initial response to the reviewer’s concerns below, asking for further clarification where appropriate. We first address the Weakness 2, as our response for Weakness 1 depends on it.
>
> ### **Weakness 2: Novelty of ROTATE compared to Previous Approaches in AHT**
>
> The novelty of ROTATE lies in the open-ended generation of novel teammate policies based on exploiting the ‘cooperative weaknesses’ of the ego agent at each iteration, while the ego agent seeks to learn to cooperate with the generated teammates.
>
> The process is open-ended (Hughes et al., 2024) because at each iteration, ROTATE always generates novel teammate policies that encourage ego agents to increase their robustness. ROTATE specifically generates novel teammates *based on* finding policies that the current ego agent cannot collaborate well with (i.e., via maximizing the cooperative regret of the ego agent). This means that ROTATE can be run for as many iterations as desired, to train an ever-more robust ego agent policy against a pool of teammates that grows at each iteration. Unlike prior two-stage methods, the generalization of an ego agent trained by ROTATE will not be limited by a fixed-size set of teammate policies.
>
> Unlike ROTATE, the minimax-Bayes approach (Villin et al., 2025) is not an open-ended AHT method since it does not continually sample novel teammate policies. As mentioned at Lines 112-113, Villin et al. (2025) explores optimizing minimax regret by learning a *sampling distribution* over a fixed and pre-specified set of teammate policies. **The teammate set is required as an input to their algorithms.** Thus, they are actually another example of a two-stage AHT method, which strongly depends on the predefined set of teammate policies during training. The minimax return and regret objectives are merely used to design a *curriculum* for sampling predefined training tasks for the ego agent.
>
> Methods that use regret to optimize how a set of tasks should be presented to a learner, are called *curator-based* methods in the UED literature. Villin et al. (2025) is a curator-based method, as are Erlebach et al. 2024 and Chaudhary et al. 2025. In contrast, methods that rely on regret to design *new* tasks for a learner are called *generator* methods (Jiang et al., 2021). **ROTATE is the first  generator-based method for AHT.** As discussed at lines 352-354, we do not compare against curator-based techniques because they are orthogonal to ROTATE, and would obfuscate the truly novel aspects of ROTATE.
>
> The above papers were cited and discussed in our paper already, but we will improve the discussion by adding an explicit discussion of ROTATE’s novelty over existing AHT methods in the Appendix, along with a figure clarifying the difference between ROTATE and curator-based methods. Additionally, we observe that there is a typo at Line 113 where the Villin et al. (2025) citation was only included as a paper considering minimax return, and mistakenly omitted from the list of AHT papers considering minimax regret. We apologize for any potential confusion and will fix this.
>
> ### **Weakness 1: Clarifying Whether ROTATE’s Per-State Regret Objective Or Teammate Generation Is More Important**
>
> We are a bit confused about what the reviewer means by this weakness, and respectfully request clarification.
>
> If the reviewer is asking for comparisons with other open-ended methods that generate teammates by optimizing alternative objectives, note that we have open-ended baselines that generate teammates by maximizing regret only on the initial state distributions (i.e., PAIRED, Rotate (per-traj)) and by minimizing the ego agent’s returns when following its current policy (i.e., Minimax).
>
> We hope our response for Weakness 2 makes the point that in ROTATE, the cooperative regret objective is inseparable from the teammate generation process, as the teammate generation is fundamentally based on maximizing cooperative regret.
>
>
> ### **Weakness 3a: Clarity of Paper**
>
> We apologize for any lack of clarity, and will seek to improve the clarity of the paper.
>
> Regarding the precise definition of the algorithm: due to lack of space in the main paper, we placed the algorithm pseudocode in Appendix A (Algorithms 1-3). Given the extra page provided by ICLR at discussion/submission time, we can move Algorithm 1 to the main paper.
>
> Regarding the definition of $\Pi^{-i}$: this term is actually introduced before Section 6, specifically at Line 229 in Section 5. However, we agree that this term should be more clearly introduced in Section 4 (Problem Formulation) and will make this improvement.
>
> Please let us know if there are any further concerns or suggestions.

---

> ### Author Response · Authors · 2025-11-18
>
> ### **Weakness 3b: Lack of Theory**
>
> We explicitly acknowledge this as a limitation of our paper in Section 8. This paper provides theoretical motivation, intuitive justification, and strong empirical evidence for the efficacy of ROTATE and the proposed per-state regret objective.
>
> We agree that exploring the theoretical properties of ROTATE is an exciting future direction, but presents substantial challenges, as sabotage itself has not been clearly theoretically formulated by the literature, and would constitute a major contribution by itself.
> We submit that the analysis presented in the current paper is in line with that of papers that are typically accepted at ICLR.
>
> ### **Question 1: Baseline Implementation**
> The only baselines we adapted were UED baselines (Minimax Return, PAIRED), which clearly do not apply to AHT problems out of the box. For the AHT baselines (FCP, BRDiv, CoMeDi), we directly compared them to ROTATE.
>
> ### **References**
>
> [1] Hughes et al. Open-Endedness is Essential for Artificial Superhuman Intelligence. ICML 2024.
>
> [2] Villin et al. A Minimax Approach to Ad Hoc Teamwork. AAMAS 2025.
>
> [3] Jiang et al. Replay-Guided Adversarial Environment Design. NeurIPS 2021.
>
> [4] Erlebach et al. RACCOON: Regret-based Adaptive Curricula for Cooperation. 2024.
>
> [5] Chaudhary et al. Improving Human-AI Coordination through Online Adversarial Training and Generative Models. 2025.

---

### Meta-Review · Area_Chair_qvi3 · 2026-01-07

**Summary:**

The reviewers raised concerns about novelty, clarity, theoretical grounding, and experimental completeness. The authors’ rebuttal clarified the novelty of open-ended teammate generation via regret minimization and addressed many presentation issues. However, limited evaluation scope and lack of theoretical guarantees remain notable limitations.

**Reviewer Concerns:**

Addressed: Novelty vs. prior work (open-ended vs. two-stage), clarity of algorithm (pseudocode to be moved), justification for missing baselines (benchmark results show they underperform).
Outstanding: Need for theoretical analysis of sabotage mitigation, evaluation in more complex settings (multi-agent, partial observability), and sensitivity analysis on objective weighting.

**Reviewer Scores:**

xw5y (4): Concerns on novelty/clarity addressed; may raise score slightly to 5.

NkUx (2): Clarity improvements and rebuttal on novelty may soften rejection, but theoretical gaps remain.

EESm (2): Similar to NkUx; weighting explanation helps, but missing baselines and theory still limit score increase.

1eRE (4): Detailed rebuttal on open-endedness definition and experimental choices likely satisfies many concerns; may raise score to 5.

---

### Decision · Program_Chairs · 2026-01-26

Reject